# Characterization of crAss-like phage isolates highlights Crassvirales genetic heterogeneity and worldwide distribution

**María Dolores Ramos-Barbero**[1,2], **Clara Gómez-Gómez**[1,2], **Laura Sala-Comorera**[1,2], **Lorena Rodríguez-Rubio** [1], **Sara Morales-Cortes**[1], **Elena Mendoza-Barberá** [1], **Gloria Vique** [1], **Daniel Toribio-Avedillo** [1], **Anicet R. Blanch** [1], **Elisenda Ballesté** [1], **Cristina Garcia-Aljaro**[1] & **Maite Muniesa** [1] ✉

Crassvirales (crAss-like phages) are an abundant group of human gut-specific bacteriophages discovered in silico. The use of crAss-like phages as human fecal indicators is proposed but the isolation of only seven cultured strains of crAss-like phages to date has greatly hindered their study. Here, we report the isolation and genetic characterization of 25 new crAss-like phages (termed crAssBcn) infecting *Bacteroides intestinalis*, belonging to the order Crassvirales, genus *Kehishuvirus* and, based on their genomic variability, classified into six species. CrAssBcn phage genomes are similar to ΦCrAss001 but show genomic and aminoacidic differences when compared to other crAss-like phages of the same family. CrAssBcn phages are detected in fecal metagenomes around the world at a higher frequency than ΦCrAss001. This study increases the known crAss-like phage isolates and their abundance and heterogeneity open the question of what member of the Crassvirales group should be selected as human fecal marker.

The discovery of crAssphage, a highly human-specific resident of the gut, has been of paramount importance, as it is the only universal marker of human fecal pollution described so far[1–4]. The first crAssphage (cross-assembly phage) is a bacteriophage or group of phages first identified by in silico studies of human fecal metagenomes[4]. This first p-crAssphage (prototypical-crAssphage) has recently been classified as the family *Intestiviridae*, species *Carjivirus communis*[5]. After the discovery of p-crAssphage, other phages (crAss-like phages), similar in their genomic architecture and sharing the same ancestor with p-crAssphage, mostly uncultured, and highly abundant in the mammalian gut and other habitats, have been discovered, all conforming a new order of Crassvirales[3,6,7]. Currently, the order Crassvirales comprises four families, 11 subfamilies, 42 genera and 73 new species (https://ictv.global/taxonomy). All of them seem to infect bacteria in the phylum Bacteroidetes and, based on the few virions isolated, they specifically infect the *Bacteroides* genus.

Given the abundance of crAss-like phage sequences identified in different metagenomes[7,8] and the increasing number of studies detecting crAssphages (or crAss-like phages) in different ecosystems by qPCR[2,9–12], it is intriguing why only seven crAss-like phages have been isolated to date. Possible explanations are the use of unsuitable host bacteria, the relatively low proportion of virulent crAss-like phages in comparison with other lytic phages infecting *Bacteroides*, or the inability of infectious crAss-like phages to generate visible plaques of lysis that allow their isolation.

Efforts to isolate crAss-like phages were not successful until four years after their discovery, when ΦCrAss001 (family *Steigviridae*, species *Kehishuvirus primarius*[5]) was isolated in *Bacteroides intestinalis*[5,13,14]. The difficulties in isolating crAss-like phages in pure culture persist, with only six other crAss-like phages isolated to date: DAC15 and DAC17 (family *Steigviridae*, species *Wulfhauvirus bangladeshii*[5]), were isolated on *Bacteroides tethaiotaomicron*[15] in

---

[1]Departament de Genètica, Microbiologia i Estadística, Universitat de Barcelona, Diagonal 643. Annex. Floor 0, E-08028 Barcelona, Spain. [2]These authors contributed equally: María Dolores Ramos-Barbero, Clara Gómez-Gómez, Laura Sala-Comorera. ✉e-mail: mmuniesa@ub.edu

2020, ΦCrAss002 (family *Intestiviridae*, species *Jahgtovirus secundus*)[5] on *Bacteroides xylanisolvens* in 2021[16], and recently, three more crAss-like phages have been isolated infecting *Bacteroides cellulosilyticus* and assigned to three new species (*Kehishuvirus winsdale* (Bc01), *Kolpuevirus frurule* (Bc03), and *Rudgehvirus redwords* (Bc11)[17]. All other reported crAss-like genomes are the result of composite assemblies and have never been propagated in pure culture in a laboratory. Consequently, there is limited information about the biology and replicative cycle of crAss-like phages. It has been established that they are virulent phages of podovirus-like morphology (short-tailed) with a relatively large (100 Kb) double-stranded circular DNA genome. Studies on the replication of crAss-like phages suggest they do not follow a common lytic cycle. Plaques of lysis of ΦCrAss001 are visible in agar overlays, but the infected liquid bacterial cultures are not cleared, suggesting that phage and host co-exist. This has been attributed to phase-variation of bacterial capsular polysaccharides, which maintains a dynamic equilibrium between phage sensitivity and resistance of *Bacteroides* cells and allows the phage and host cell to multiply in parallel[18]. ΦCrAss002 does not form plaques or spots on lawns of sensitive cells, nor does it lyse liquid cultures, even at high titers, and the phage only propagates if co-cultured with the host for a minimum of 3–5 days[16]. Phages DAC15 and DAC17 might form plaques of lysis but they propagate poorly in the *B. thetaiotaomicron* wild type strain and more readily in mutant strains with single capsular polysaccharides[15].

Metagenomic analysis has enabled the identification of many crAss-like phage genome sequences, and multicenter investigations have concluded that crAss-like phages, especially p-crAssphage, are highly abundant in human feces and have a global distribution[8]. As a result, p-crAssphage, and in general crAss-like phages, have been proposed as universal indicators of human fecal pollution[10,12,19,20], despite the genetic variability within the Crassvirales order[8]. qPCR analysis has also revealed crAss-like phages in samples with animal fecal contamination, although at much lower levels than in human feces[12]. It has also been suggested that crAss-like phages persist in the environment for longer than *Escherichia coli* and comparably with somatic coliphages[21], an essential attribute of an effective fecal indicator.

Nevertheless, the molecular methods used so far for crAss-like phages detection are limited in that they cannot distinguish between the target and non-infecting or inactivated viruses. Moreover, the abundance and persistence of crAss-like phages in the human gut has not yet been explained mechanistically and, as suggested previously, the phage-host relationship can only be properly studied with isolated phage-host pairs[16].

In this work, we present the isolation and characterization of 25 new virulent crAss-like phages infecting *B. intestinalis* that represents a step forward in the analysis of this ubiquitous human-specific phage group.

## Results

### Isolation of crAss-like phages

CrAss-like phages were searched in 12 sewage samples from five wastewater treatment plants using three different qPCR assays, the one designed from ΦCrAss001 genome in this study and two previously described[12,19]. Depending on the qPCR assay employed and the samples, values in wastewater ranged from $10^3$–$10^6$ gene copies (GC)/ml of crAss-like phages. If considered the qPCR assay designed from ΦCrAss001, the one finally used, $10^3$–$10^5$ gene copies (GC)/ml of crAss-like phages were detected (Supplementary Table 1). Assuming that one GC corresponds to one phage particle, the analyzed waters contained $10^3$ to $10^5$ viral particles/ml.

Different qPCR assays revealed the presence of crAss-like phages but were unable to determine whether they were infectious phages that could be isolated or propagated, a task that has proven challenging to date[16,18]. The three different qPCR assays were tested in enrichment cultures of different strains of *Bacteroides* (Supplementary Table 2), to identify the most suitable host for the propagation of crAss-like phages present in the wastewater samples and thus facilitate their isolation. Three consecutive propagation steps were performed, and the abundance of crAss-like phages was monitored with the different qPCR assays after each step. *B. intestinalis* was the only host that generated a notable increase in the number of crAss-like phage particles detected with the qPCR designed for ΦCrAss001, after the second or third propagation step in nine of the 12 samples analyzed (Supplementary Table 1). Phage suspensions obtained directly from wastewater samples or after propagation in these nine samples were diluted and plated. Plaques of lysis (plaque-forming units: pfu) generated by crAss-like phages were detected by plaque blot hybridization using the CrAss1-ORF46 probe. A clear hybridization signal was obtained only from the propagated suspensions, which facilitated the enumeration of the tiny plaques (Supplementary Fig. 1). On average, 20–30% of the plaques visualized on the soft agar overlay showed a positive signal for the CrAss1-ORF46 probe, although the proportion may have been higher, given the difficulty in visualizing the smallest plaques (Supplementary Fig. 1). Finally, 1–5 positive plaques from each wastewater sample, well separated from the other plaques and showing a clear hybridization signal, were isolated, purified, and confirmed to be crAss-like phages by PCR. Twenty-five phages (named crAssBcn phages, from ΦCrAssBcn1 to ΦCrAssBcn25) from samples of the different wastewater plants and collected in different dates were randomly selected for further characterization (Supplementary Table 1).

### Characterization of the new crAss-like phages

All crAssBcn phages showed a podovirus-like morphology with head diameters of $77 \pm 0.8$ nm and a short tail of $40 \pm 0.9$ nm with a characteristic trident shape (Fig. 1). Despite the different origins of the crAss-like phages analyzed, no morphological differences between them were observed.

When propagating in liquid culture, maximum titers of crAssBcn phages were obtained after 24 h of propagation reaching values of up to $2 \times 10^9$ pfu/ml, with increases of 2.5–4.0 log units between 0 and 24 h (Supplementary Fig. 2a). An initial explosion in the infectious process at 2 h of incubation, with an increase of 1 log unit, was followed by a lower increase at 2–4 h. A second substantial increase of 1 log unit was observed at 4–6 h of incubation, followed by a slight increase at 6–9 h. From 9 to 24 h, the number of infectious crAssBcn phages remained constant, all reaching values of $4 \times 10^7$ – $2 \times 10^9$ pfu/ml. Although differences were observed, the mean propagation curve of the 25 crAssBcn phages (dotted blue line in Supplementary Fig. 2a) did not differ significantly from that of ΦCrAss001 (red line) (Wilcoxon matched pairs test, $p = 0.125$). Likewise, no differences were observed between individual crAssBcn phages (grey lines) ($p > 0.05$).

The average burst size of the crAssBcn phages was calculated as $64.6 \pm 16$ phages per infected cell. However, the crAssBcn phages did not affect the host strain growth. The infected cultures were not cleared during propagation, and instead the number of host cells increased throughout, reaching an optical density (OD) at 600 nm close to 1.0 at 24 h (Supplementary Fig. 2b). A similar pattern was revealed by measuring the number of culturable cells, which fluctuated until 9 h after infection and reached the highest concentration at 24 h (on average $4 \times 10^8$ cfu/ml). The concentration of the uninfected *B. intestinalis* control was only slightly higher (on average $8.8 \times 10^8$ cfu/ml) than the infected *B. intestinalis* cultures after 24 h (Supplementary Fig. 2b).

### Intergenomic comparison of the 25 crAssBcn phages and other crAss-like phages

Sequencing of the 25 isolates allowed the identification of 24 dsDNA complete genomes and one draft genome (ΦCrAssBcn25). The 24 complete phage genomes ranged from 97,685 to 103,497 bp in size

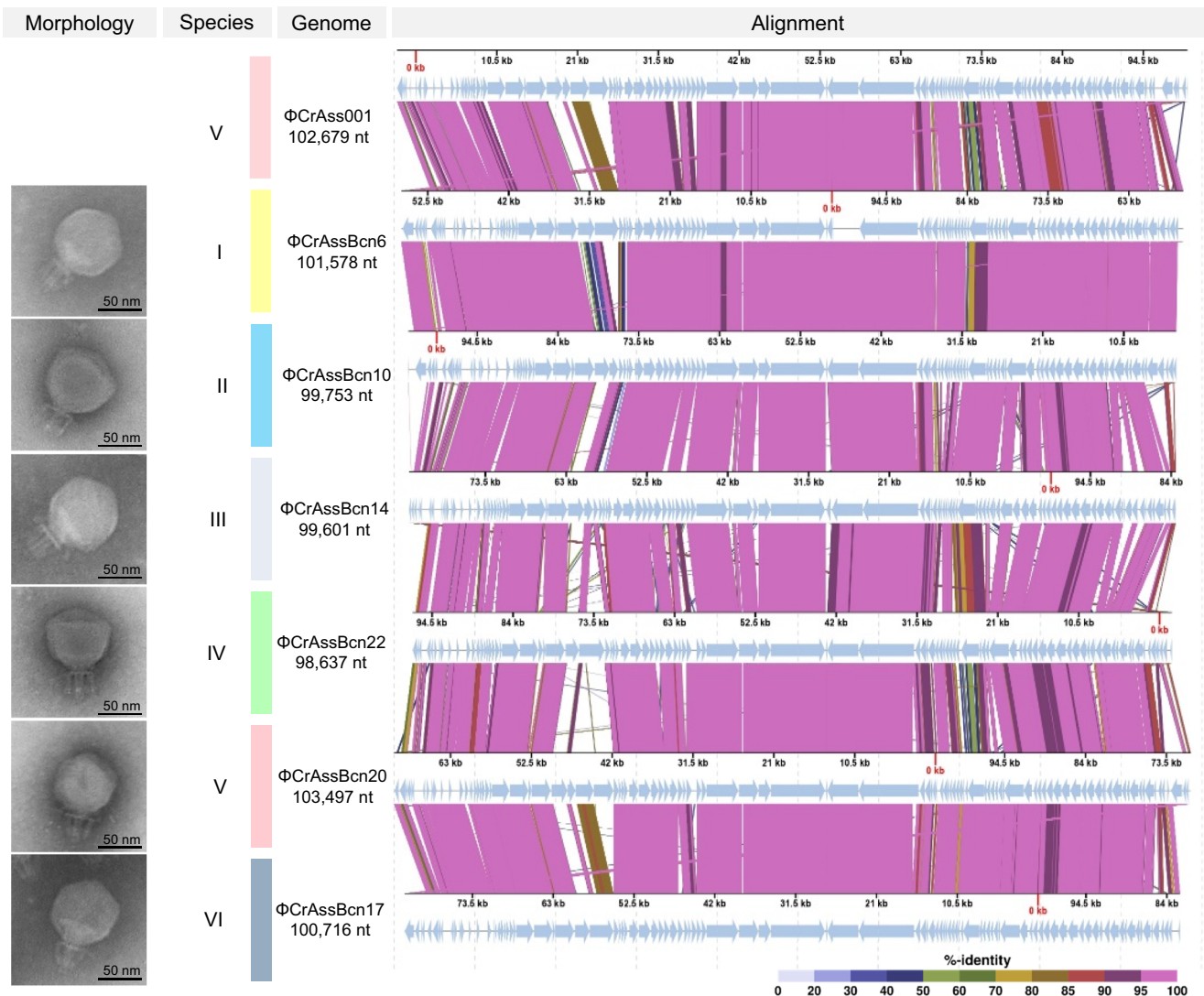

**Fig. 1 | Comparison of the six representative crAssBcn phages of each species and ΦCrAss001.** Electron micrographs and CDS alignment of the genomes of six crAssBcn phages representative of the six species (groups I-VI). Light blue arrows indicate the ORFs. The color of the bands between the seven genetic maps shows the percentage of identity between each sequence as indicated in the legend at the bottom of the figure.

and contained 101 to 114 ORFs (Supplementary Table 3, Supplementary Fig. 3).

Comparison of the 24 crAssBcn phages showing complete genomes (incomplete phage ΦCrAssBcn25 genome was excluded from the following analyses) revealed that none shared an identical nucleotide sequence (Fig. 2, Supplementary Fig. 3) and they all also differed at the amino acid level confirming that technically none had identical genomes. The phage with the largest complete genome within each species, according to the grouping suggested by VIRIDIC (in nt), was selected as representative of that species(Figs. 1, and 2) Based on the recently revised taxonomy[5], all the crAssBcn phages were classified within the order Crassvirales, family *Steigviridae*, subfamily *Asinivirinae*, and genus *Kehishuvirus*.

According to the International Committee for the Taxonomy of Viruses[22], the main species demarcation criterion for bacterial and archaeal viruses is currently set at a genome sequence identity of 95% over 85% of the complete genome. Using this criterion, the group of 24 crAssBcn phages could be divided in six different species, (I to VI) (Fig. 2, Supplementary Table 3), each comprising from 1 to 11 phage genotypes. Species I was the most numerous (11 phages), representing 44% of the isolated crAssBcn phages. Only ΦCrAssBcn20 (species group V) showed a genome sequence identity of > 95% with

ΦCrAss001, being a member of the same species (*Kehishuvirus primarius*) (Fig. 2).

Functional annotation of all crAss-like phages is presented in Supplementary Data 1. Exemplifying the crAssBcn phages, the genetic map of the isolate ΦCrAssBcn6, representative of species I, shows the distribution of 105 ORFs (Supplementary Fig. 4). All the crAssBcn phages were probably virulent, as genes related to lysogeny (i.e., integrase, excisionase, and lysogenic module genes) were not detected. Additionally, the high number of ORF genes related to metabolism and replication (Supplementary Fig. 4) suggests the phages had a high replicative potential, which is characteristic of virulent phages.

To explore the phylogenetic relationship between crAssBcn phages with their closest relatives, we performed a phylogenetic analysis including as query the 25 crAssBcn phages and the 15 uncultured crAss-like phages assigned within the 15 species of the family *Steigviridae*[6] (in Supplementary Table 4). The proteomic tree showed that the 25 crAssBcn phages clustered with different levels of similarity on the same branch, where ΦCrAss001 was also clustered. In contrast, crAss-like phage genomes of species assigned to *Steigviridae* family were close but in different branches (Fig. 3).

Alignment of the ΦCrAssBcn6 genome, the representative of species I, with the genomes of the 15 phages of the *Steigviridae* family

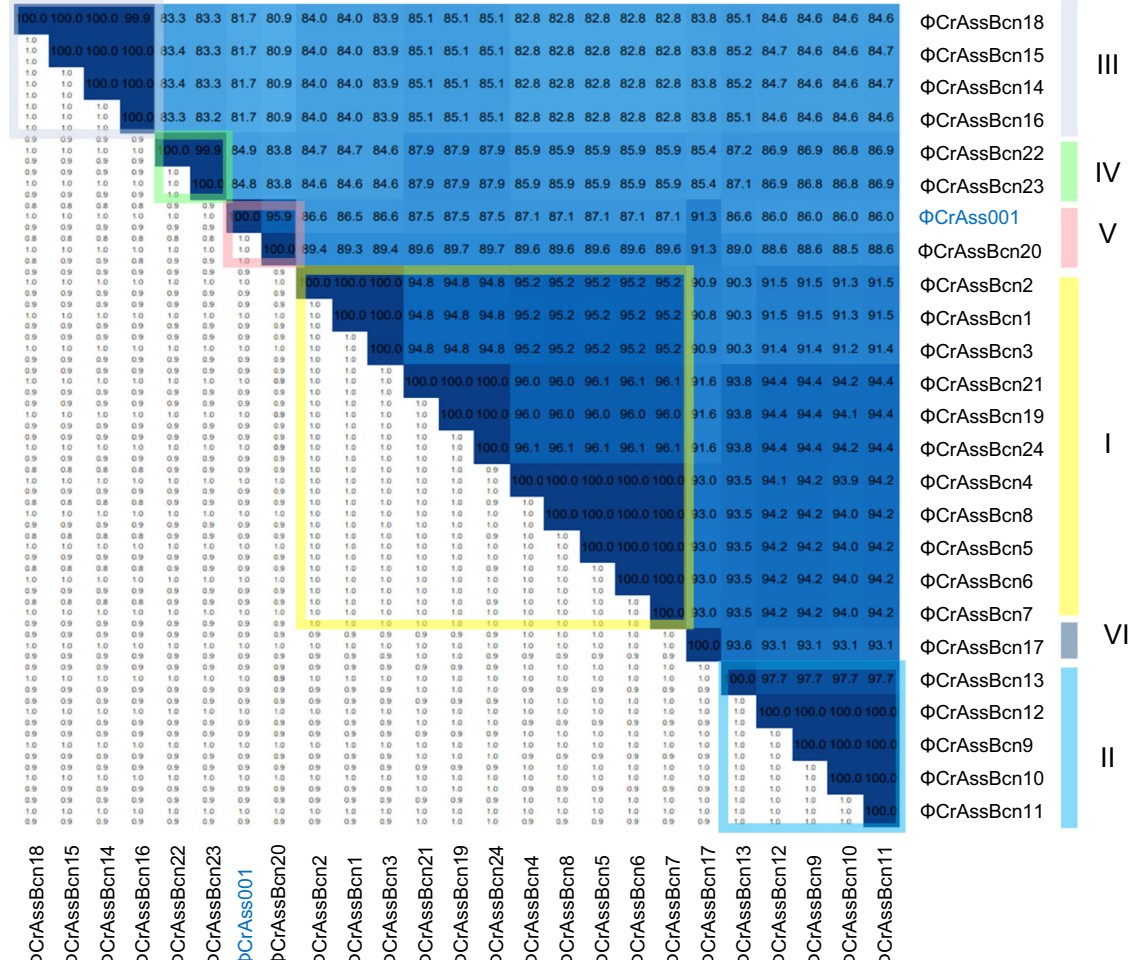

**Fig. 2 | Intergenomic comparison of the 24 complete crAssBcn phages genomes and their classification into species.** Heatmap generated by VIRIDIC shows the intergenomic similarity values (right half) and alignment indicators (left half). The percent identity between two genomes was determined by BLASTn, integrating intergenomic similarity values with data on genome lengths and aligned genome fractions. In the right half, the more closely related the genomes, the darker the color. The numbers inside the map represent the identity values for each genome pair, rounded to the first decimal. In the left half, the aligned genome length is indicated, with darker colors corresponding to smaller aligned genome fractions. On the right side of the heatmap, the six species (I to VI) in which each virus is classified are indicated in colored squares.

allowed to visualize the low identity at the genome level (Supplementary Fig. 5), therefore crAssBcn phages cannot be assigned to any of the previously described species, except ΦCrAssBcn20, as indicated above.

**Variability within the crAssBcn phages**

When comparing the genomes of the 24 crAssBcn phages (ΦCrAssBcn25 was excluded), the first observation was that some ORFs were detected in some, but not in all, of them (Supplementary Table 5). Some ORFs were absent in 100 to 8% of the phage genomes. Among them, many encoded hypothetical proteins, endonucleases or tail fibers (Supplementary Table 5).

Considering those ORFs present in all or most of the genomes, we aimed to identify those showing the highest variability among the 24 crAssBcn phage genomes. To avoid generating an excessively large data matrix, which would have been difficult to interpret, only the genomes of the six representative phages were used to ascertain the differences between the *ca* 100 ORFs of each phage. Using a reciprocal best-match analysis, the comparison revealed the ORFs with the lowest % of amino acid identity (AAI) and hence with the highest variability in the phage genomes (Supplementary Fig. 6). Seven ORF with the lowest % AAI were selected and aligned (Figs. 4 and 5) to illustrate the degree of variability observed. According to

the last annotation of ΦCrAss001 (released in January 2023) and a recent cryoEM identification of ΦCrAss001 structural proteins[14], the most variable ORFs (AAI close to 50%) encode a tail spike, a tail protein, a head protein, a holin, a HNH-endonuclease, a hypothetical protein (Fig. 4), and a DNA polymerase I (Fig. 5). To a lower extent, a certain variability in thymidylate synthases, C-type lectin, and RNA polymerases was also observed (See Supplementary Data 2 for detailed information). The phylogenetic trees generated with these highly variable proteins were constructed with the sequences of crAssBcn phages and the closest hits showing a genome coverage larger than 50% and an identity over 50% or, in case that no sequences accomplished these criteria, the ten best hits in the databases were used (datasets are included in Supplementary Data 3). The tail proteins and the HNH-endonucleases of all crAssBcn phages derive from a single common ancestor, although some are not present in all the crAssBcn phages (below the detection limit) (Fig. 4). The crAssBcn phages head proteins derive from a single cluster that later shows a dichotomy in two different variants, a dichotomy that includes the head protein of phages ΦCrAssBcn20 and ΦCrAss001, that differ even though both phages belong to the same species V (Fig. 4). In contrast, the tail spike proteins, holins and the hypothetical protein of crAssBcn phages are located in different clusters, similarly to other crAss-like phages of the databases (Fig. 4).

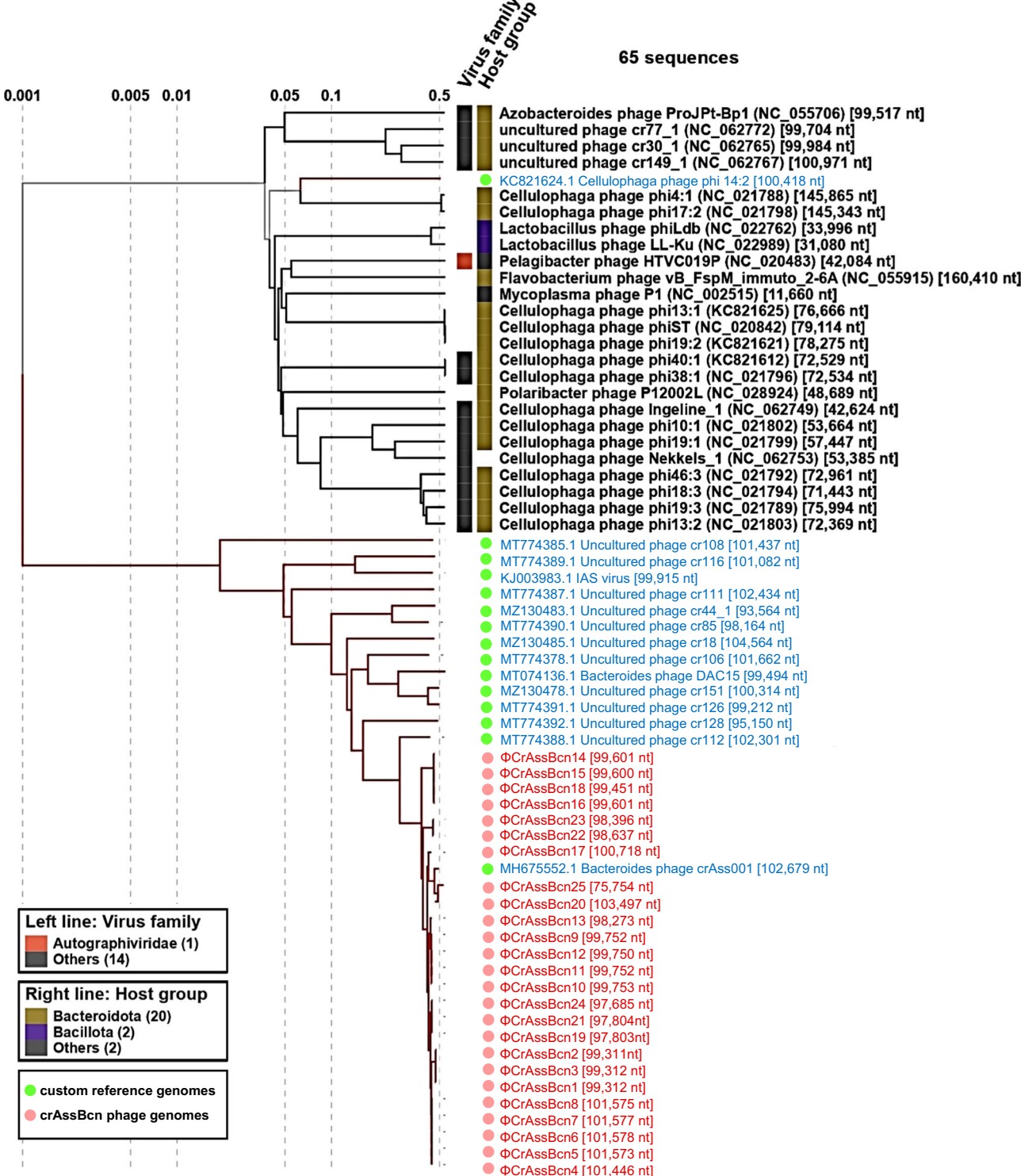

**Fig. 3 | Phylogenetic relationship of the crAssBcn phages.** Viral proteomic tree constructed with Viptree, where the 25 crAssBcn phages (in red) and 15 crAss-like phages of the *Steigviridae* family (in blue) were included as a query, showing the phylogenetic relationship of each phage with its closest relatives.

Analysis of the DNA polymerase A of the representative crAssBcn phages species I-VI (Fig. 5) showed that all six species encoded a family A DNA polymerase I. However, the alignment and % AAI for the polymerases confirmed two different types with a 45% of AAI between them; one of 706 aa, identified in species V and VI (phagesΦ-CrAssBcn20, ΦCrAssBcn17) coincident with the ΦCrAss001 DNA polymerase, and another one of 775 aa in 22 phages in species I to IV (Fig. 5). Both types of polymerases show hits in the databases (Fig. 5),

as the polymerase type of species V-VI is present in other phage genomes assigned to *Steigviridae* family (for example GenBank accession number YP_010111476.1), or to polymerases detected in metagenomes or in sequences previously reported by Yutin et al. [7] (Fig. 5). The polymerase variant of species I-IV showed the closest hit (97.4%) in a metagenome without species assigned (MAG sequences in Fig. 5), and one of the closest phage genomes was that of the uncultured phage cr106_1, assigned to species *Mahstovirus faecalis* (accession N°

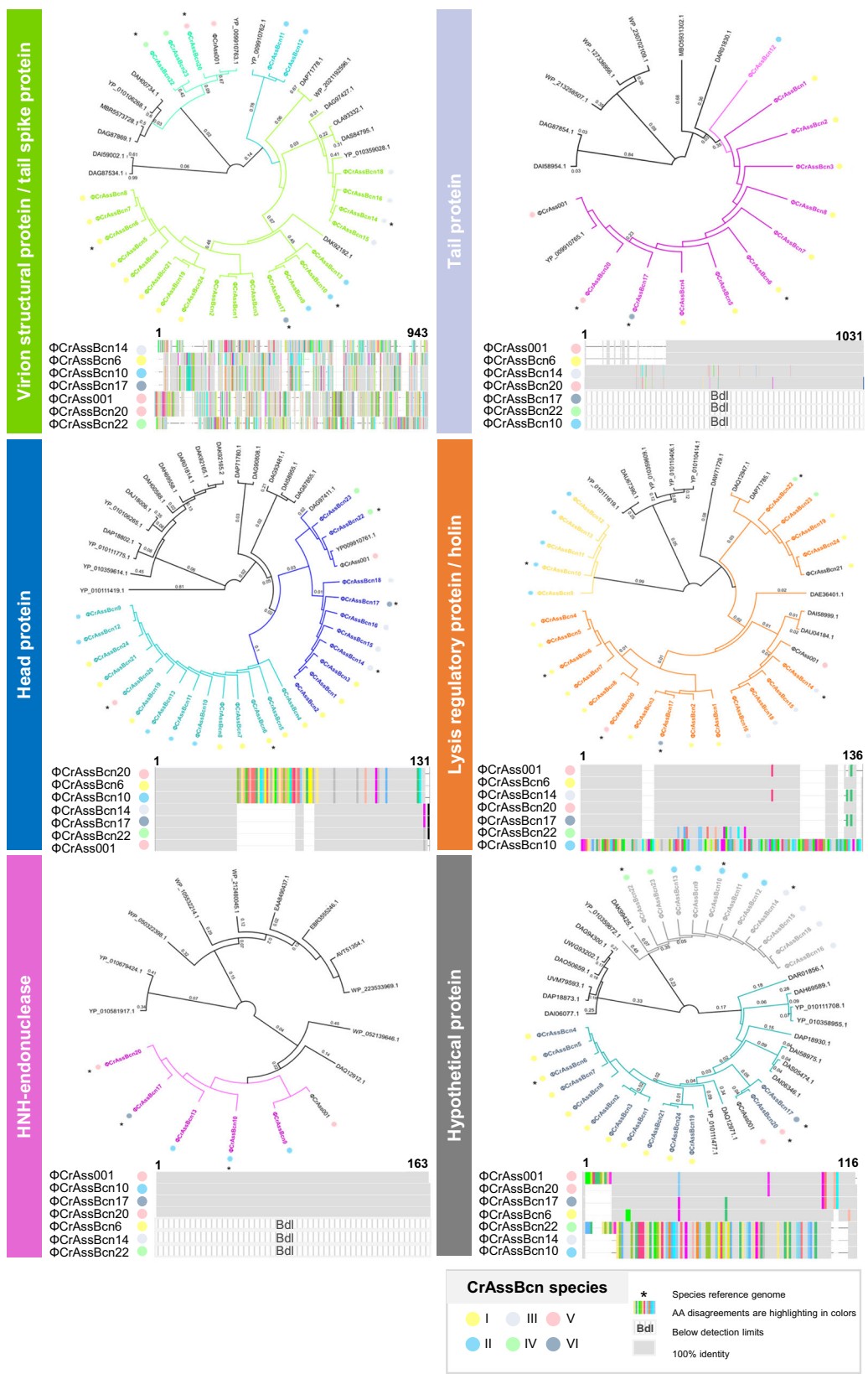

**Fig. 4 | Comparison of the most variable ORFs of crAssBcn phages.** Phylogenetic tree and multialignment of the amino acid sequences of the different variable ORFs of the CrAssBcn phages and ΦCrAss001. In the phylogenetic tree, the 24 crAssBcn phage sequences are displayed in coloured lines, while sequences from the databases are displayed in black lines. Only branch labels over 0.001 are shown. The multialignment of the different ORFs was constructed with the representative genomes of the six species where 100% identity is displayed in grey bars and aminoacid disagreements are shown in coloured bars. The color code for the crAssBcn species is indicated in the legend. Those proteins not detected in a given species are indicated a Bdl (below detection limit) and consequently are not present in the phylogenetic tree. The asterisk indicates the species reference genome.

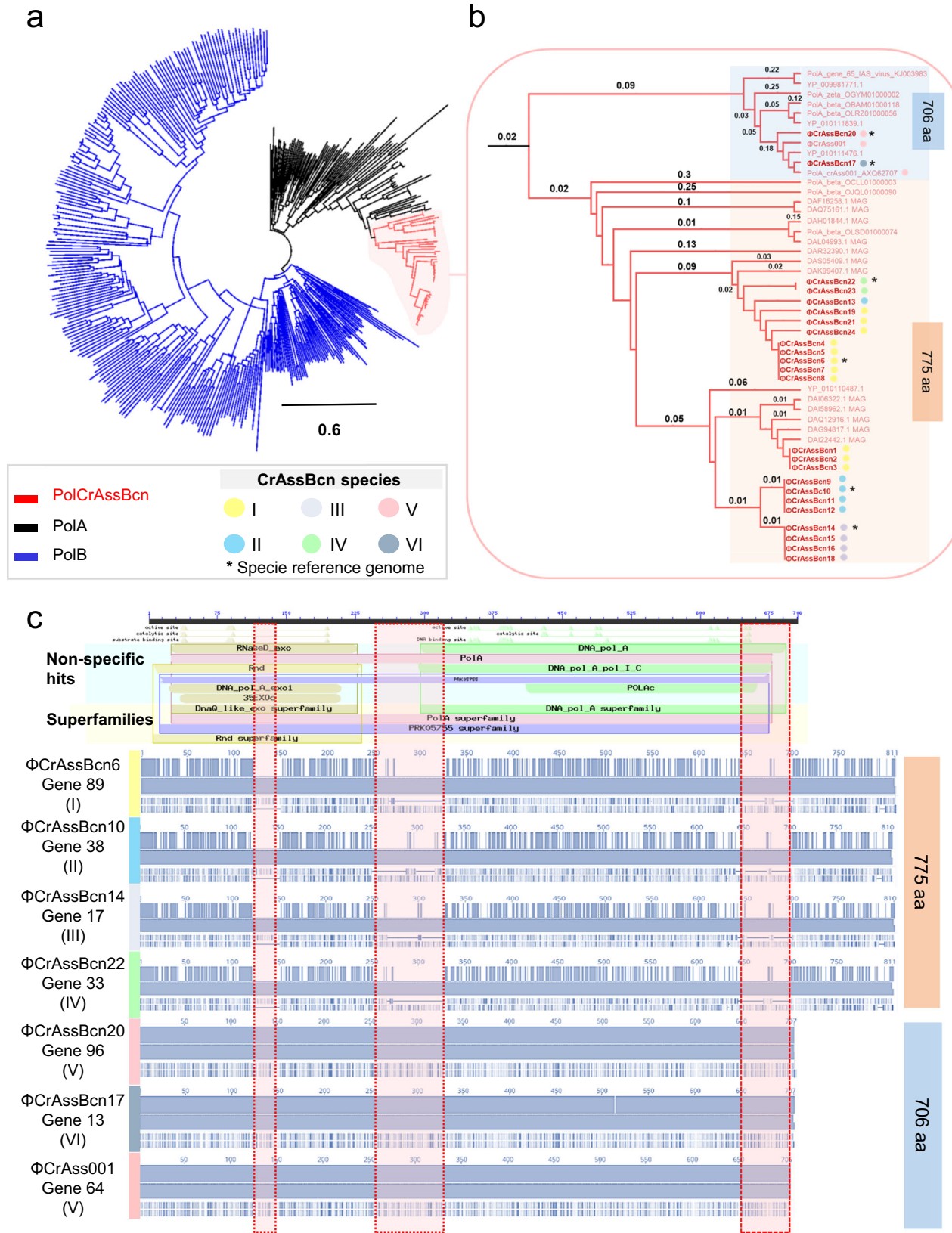

YP_010110487.1) (Fig. 5 and Supplementary Data 3), but with only a 79.3% of identity. Phylogenetic tree (Fig. 5) shows that PolA evolved from PolB, that both variants of the PolA of crAssBcn phages seem to have evolved from a common PolA ancestor and that from this point, they have diverged in the two different types observed. The type with 775 aa appears to be also the most abundant in the crAssBcn phages

and is also the most represented among the crAss-like phage sequences available in the databases.

The archetypical structure of the polymerase A should comprise three domains to sustain its activity. The polymerase active site with three subdomains: fingers (which bind an incoming nucleotide and interact with the single-stranded template), palm (which harbors the

**Fig. 5 | Comparison of the polymerase of the six species of crAssBcn phages. a.** hylogenetic tree constructed with the polA of crAssBcn phage, other PolA and PolB proteins available in databases and reported in crass-like phages by Yutin et al., (2021). **b** Detail of the branch of the phylogenetic tree were crAssBcn phages are located showing the two separated branches containing the 706aa and the 775aa family A polymerases. **c** Multiallignment of PolA of the six species of crAssBcn phages. Upper part shows the polymerase A domains using the polymerase A gene of phage ΦCrAss001 as reference. From top to bottom this chart shows first the

length of the protein. Small triangles indicate the aminoacids involved in conserved active, catalytic and DNA binding sites of each domain. Colored bars show the closest hits found in the conserved domain database (CCD) for each domain, these can be specific hits (with a high confident association) or non-specific and the superfamily to which the highest-ranking hit belongs. In the bottom part, the alignment of the polymerases of the six species (groups I-VI) and ΦCrAss001. Vertical red shadows bands highlight the gaps observed between the 775aa and the 706aa polA.

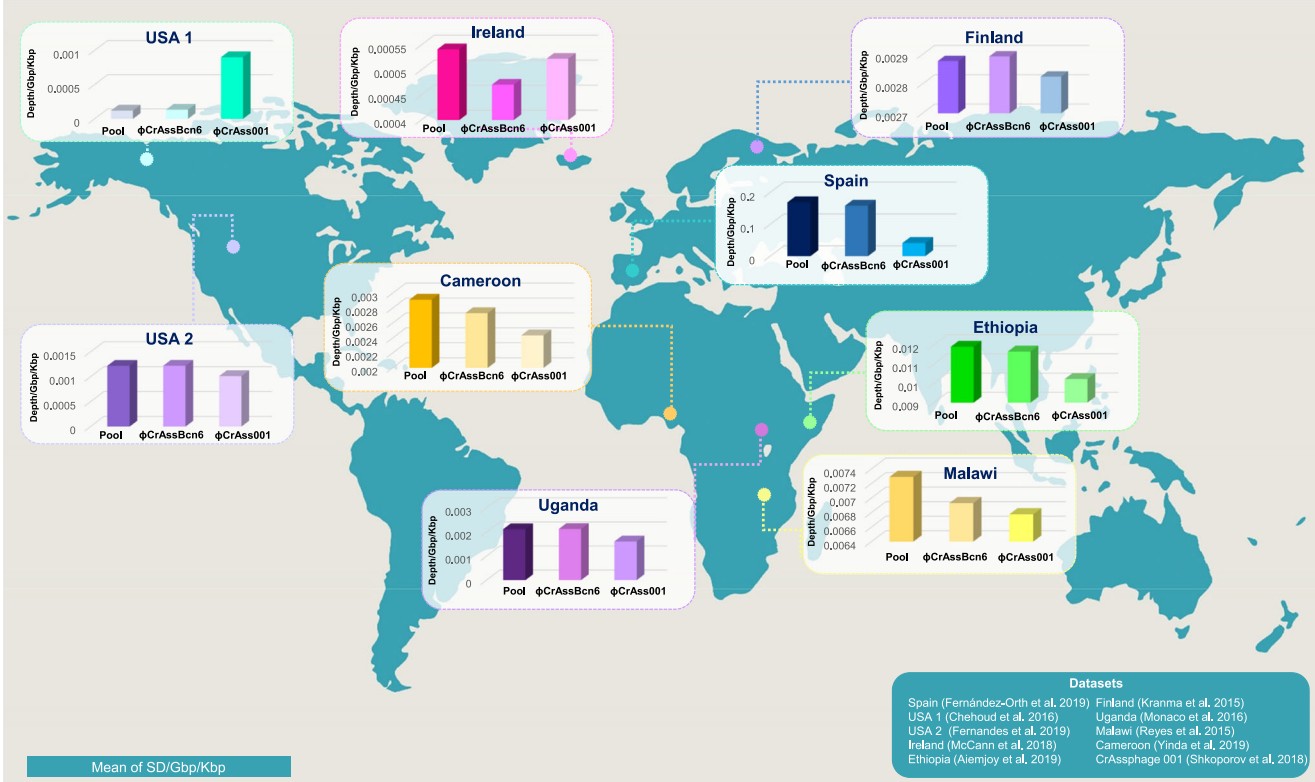

**Fig. 6 | Geographical distribution of the new crAssBcn phages.** The presence and abundance of the crAssBcn phages in different gut metagenomes from around the globe was analyzed by fragment recruitment of the pool of 25 crAssBcn phages (Pool), the representative ΦcrAssBcn6 and the model phage ΦCrAss001. Only metagenomes with positive recruitments are depicted, which are from the USA

(USA 1 and 2), Finland, Spain, Ireland, Uganda, Malawi, Ethiopia, and Cameroon. Only mapped sequences with coverage of at least 70 % and best-hit reads were considered. Additionally, relative abundances of studied crAssphages were normalized by metagenome size (Gbp) and phage length (Kbp). World map figure in the background has been obtained from Designspot/Freepik.

catalytic amino acid residues and also binds an incoming dNTP) and thumb (which binds double-stranded DNA), and the additional associated domains for 3′–5′ exonucleolytic proofreading.

Both polymerases showed a hit for 3′–5′ exonuclease as well as a hit for the polymerase A domain (Supplementary Fig 7). Despite the observed differences in the 3′–5′ exonuclease and polymerase A domains between the two types (Fig. 5) the active sites of both domains appeared conserved.

The genes immediately upstream and downstream the polymerase, annotated as muconolactone isomerase and hypothetical protein, respectively, were also analyzed. While the ORF upstream the polymerase is conserved among the crAssBcn phages (Supplementary Data 2), the ORF immediate downstream the polymerase shows a high degree of variability among them, similar to the variability found in the ORF encoding the polymerase. This highly variable ORF was selected in our previous analysis (Hypothetical protein in Fig. 4), and encodes two types of proteins, one in species II, III and IV and another one in species I, V and VI. The nucleotide and aminoacidic sequences and the domain analysis did not, however, allow the identification of these proteins.

### Biogeographical distribution of crAssBcn phages

In the assessment of the geographical distribution of crAssBcn phages, a positive hit was obtained in 51% of the metagenomes analyzed (Supplementary Data 3), including datasets from Ireland, USA (two datasets), Spain, Uganda, Malawi, Cameroon, Ethiopia and Finland (Fig. 6 and Supplementary Fig. 8). The pool of crAssBcn phages was more represented than ΦCrAss001, except in the USA-1 metagenome, as was ΦcrAssBcn6, with the exception of USA-1 and Ireland (Fig. 6). The largest recruitment and therefore the highest relative abundance corresponded to metagenomes from Spain, followed by Ethiopia and Malawi (Supplementary Fig. 8).

### Discussion

The propagation strategy resulted in the successful isolation of 25 new crAss-like phages from wastewater. The abundance of crAss-like phages in wastewater using the different qPCR assays[12] differed because the assays target different crAss-like phage genomes. In fact, two of them[12,19] did not detect crAssBcn phages.

In this study, using different environmental and clinical isolates of *Bacteroides* and three different qPCR assays, only *B. intestinalis* allowed

efficient propagation of crAss-like phages detected by the qPCR assay targeting ΦCrAss001. Although other species of *Bacteroides* have been described as crAss-like phage hosts[15,16,23], *B. intestinalis* is also the host of ΦCrAss001, suggesting that this family comprises phages infecting this *Bacteroides* species. In their natural habitat, it is possible that these phages may infect other hosts[13] and it is also possible that other crAss-like phages infecting *B. intestinalis* would have been detected using a different set of primers and probes.

Attempts to isolate crAss-like phages from plaques of lysis generated directly from wastewater samples (without propagation) were unsuccessful, probably because the proportion of crAss-like phages among the entire phage pool infecting *Bacteroides* strains was too low and felt below the detection limit of the double agar layer technique after the dilutions performed to obtain separated plaques. Another factor hindering the isolation of crAss-like phages was the poor visibility of the generated plaques. The use of this *B. intestinalis* strain allowed the preferential propagation of crAss-like phages among other phages in the original sample, increasing their proportion and allowing their isolation. However, the use of a specific qPCR targeting ΦCrAss001 and this particular host promoted the isolation of phages related to ΦCrAss001. While other strategies based on the sequencing of the viruses present in bacterial cultures supernatants would have allowed the identification of other crAss-like phage genomes, they would not have guaranteed their isolation.

The average burst size of the crAssBcn phages was of 64.6 ± 16 phages per infected cell. In contrast, ΦCrAss001 shows a burst size of 2.5 phages per infected cell[13], although, microscopy observations indicated progeny counts of >50 phages per cell[13], closer to the burst size of crAssBcn phages. In addition, the propagation curves of crAssBcn phages consistently show a sequence of smaller bursts during the 9 h of incubation, starting at 2 h after infection and after 4 h and 6 h. Phage propagation did not seem to be detrimental for the growth of the strain, suggesting that the host population was maintained through a dynamic phage-host interplay, involving at least a fraction of the bacteria. These observations, in agreement with previous studies[13,18] indicate that crAssBcn phages are not temperate, and as lysogeny is ruled out, the persistence of the host bacteria during phage propagation may be due to a carrier-state infection, pseudolysogeny, or phase variation-based resistance involving receptor variation[24], alone or in combination[18]. Phase-variable systems that promote resistance, show a dynamic of spontaneous reversion of some resistant cells to a susceptible state. The sensible fraction of the bacterial population could generate enough susceptible cells to propagate the phage progeny while the resistant fraction maintains the bacterial population[24].

Samples of wastewater were obtained from different origins to increase the possibility of variation in any isolated phages. Although all belonged to the same genus and morphological group, the crAssBcn phages differed at the genomic and the protein level between them and when compared with crAss-like genomes of the *Kehishuvirus* genus.

The ORFs with the highest variability AAI (close to 50%) encoded variable proteins such as C-type lectin, which facilitates phage adaptation to the host[25], or thymidylate synthase genes, used by phages to acquire metabolites for DNA replication independently of the host. The variability of phage-hypermodified thymidylate synthases would expand the range of nucleotide modifications used to counteract bacterial defense systems[26].

A high level of variability was found in endonucleases, which are involved in DNA recombination, repair, and packaging, and have been proposed as a source of phage genetic diversity through positive selection pressure[4,27]. In addition, tail spikes were among the most variable genes, in accordance with the strong selective pressure these structures are under[14,28,29]. However, it was not expected to be so high in phages infecting the same host strain and presumably using the

same receptor. A possible explanation for this observation is that capsular polysaccharides or other cell surface proteins regulated by phase variation act as highly dynamic crAss-like phage receptors[17,18]. In such a *Red Queen* scenario, the phage tail protein is adapting to constantly evolving target cell surface proteins[29]. This would suggest an evolution from a single ancestor, like the one observed for the tail protein. It is also possible that different tail spikes can attach to the same receptor or that spike variability allows phages to infect the bacteria through different receptors. This might have been the case of the variable tail spikes, located in different branches of the tree, plausibly as a consequence of one or more speciation events. Similarly, different variants located in different branches were observed for the hypothetical protein and the holin, with these variants showing hits with other phage sequences in the databases, reinforcing the hypothesis of speciation through orthologous gene exchange[30]. In a lower degree, the head protein can be traced back to a common ancestor, but have diverged from there in two different types, apparently after an integration event affecting the middle part of the protein.

Another unexpected result was the difference in the AAI (less than 50%) of the DNA polymerase A gene of species V and VI compared to the other species, considering they all belong to the same genus and infect the same host. Evolutionary switches in DNA polymerase from type A to B have been described in different families of crAss-like phages[7], as well as the absence of both enzyme types[31], but this is not the case of crAssBcn phages, that encoded only a family A DNA polymerase I. Polymerase A is present in the 25% of the dsDNA phages described[32], including crAss-like phages of the *Steigviridae* family[7]. However, differences in the DNA polymerase I of species V and VI were observed compared to the other species, these differences affect 3′–5′ exonuclease and polymerase A motifs. DNA polymerases of family A have replicative and repair activities. A 3′–5′ exonuclease activity that provides proofreading and repairs the mispaired nucleotides and a 5′–3′ DNA polymerase activity for DNA synthesis. Like in phage T7 and other organisms, the DNA polymerases of crAssBcn phages have a bipartite architecture with a C-terminal polymerase and N-terminal 3′-5′ exonuclease domains that are encoded in a single gene[33]. This structure supports the dual replicative and repair activities of the polymerase A in crAssBcn phages[33].

According to the propagation curves, the replication of the different crAssBcn phages did not show differences regardless the type PolA encoded and, in addition, the different domains in the polymerase seem to be active. We concluded that the differences in the two DNA polymerases do not affect their activity and that differences among phages is the result of evolutive or recombination events, resulting in the co-existence of different polymerases with the same function. Considering that similar sequences of both polymerase types can be found in other crAss-like phages, the possibility of recombination events between phages, or between bacterial hosts and phages seem plausible[7,34]. Moreover, the highly variable ORF located immediately downstream the polymerase gene might have been mobilized in the same module as the polymerase gene during a previous recombination event. The idea, proposed by Botstein in 1980[35], is that phages evolve by shuffling interchangeable functional modules, and that selection acts on those modules facilitating the emergence of new mosaic genotypes that provide advantages in each niche. The phylogenetic analysis suggests a common PolA ancestor evolved in an early stage from PolB. From this ancestor, the two PolA variants of this study diverged showing a dichotomy. Albeit the two PolA variants are found in different crAss-like phages, the PolA of 775 aa is more represented than the smaller one, suggesting it could be more successfully spread. The divergence between the two PolA types might result in a non-orthologous gene displacement, as it was previously proposed to explain the evolution of viral DNA polymerases[34,36,37]. Maintaining diversity in the replicative modules might be beneficial for phages as it would give them the possibility of adapting to changes in the process

of coevolution between bacteria and phages, or of escaping bacterial defenses[38].

DNA polymerase I has been proposed as a signature gene for the identification of viral sequences in metagenomic samples, the reconstruction of phylogenetic trees, and the exploration of viral diversity and evolution in different habitats[39,40]. However, the variability of DNA polymerase observed in the present study suggests that it may not be a suitable target for the universal detection of crAss-like phages[41], and that other more conserved genes, such as terminase[3], should be used in preference. Conversely, the RNA polymerase, a hotspot of positive selection and thus a variable region[29], was found to be only slightly variable among the crAssBcn phages.

When compared to other crAss-like phages of the *Kehishuvirus* genus, the crAssBcn phages did not match any of them, with the exception of ΦcrAssBcn20 classified as belonging to the same species (*primarius*) as ΦCrAss001, although differences can be observed, for instance in the head protein (Fig. 4). In agreement with previous studies[7], this observation highlights the need to use crAss-like phage genomes with a sufficient identity at a representative length of their genomes or genomes obtained from isolated crAss-like phages for a correct classification. Many of the publicly available complete crAssphage genome sequences have been cross-assembled from multiple individuals or represent pooled samples[29], which may introduce significant sequence variation and confound variant analysis. In other cases, crAss-like phage identification is based only on a genome fragment (sometimes very short).

The high prevalence of crAssBcn phages in global gut metagenomes, showing a higher representation than their closest relative ΦCrAss001, was striking. Phylogeographical analysis of crAss-like phages has revealed local clusters of genetically similar phages although a geographically widespread phage strain has been reported[8]. According to this, crAssBcn phages are highly abundant in Spain but are also unexpectedly detected in other countries. The conservation of such ubiquitous strains, including the crAssBcn phages, can be attributed to recent human migration movements, high fitness, or environmental stability[8].

Due to its ubiquity in the global human population and abundance in fecally polluted water, p-crAssphage has been proposed as a human fecal indicator. Other Crassvirales are also highly abundant in the mammalian gut, particularly, in humans[3,8,31,42]. However, before other Crassvirales can be employed as human fecal markers, several issues need to be addressed. To date, crAssphage has been largely targeted in many reports, and reported as if it were a single virus, even though there is a great heterogeneity within the Crassvirales order. The few virions isolated until now are not p-crAssphage, infect different *Bacteroides* strains, and have low synteny and very low homology[16]. Assuming the diversity of crAss-like phages, the most abundant and widespread should be the best candidate to be selected as a potential fecal indicator. However, to establish which member of the Crassvirales order is the most representative on a global level will require extensive research. The detection of crAssBcn phages in metagenomes of different countries is another indication that new crAss-like phages, as yet unidentified, could be more abundant and ubiquitous than those described so far. Moreover, the 25 crAssBcn isolates illustrate crAss-like phages diversity within the same genus, which may hinder comprehensive screenings of crAssphage in contaminated water samples. The diversity of crAss-like phages should be taken under consideration particularly when designing new qPCR assays other than those already described[19]; for instance, some qPCR assays being used for crAssphage detection[8,12,19,43] would fail to detect crAssBcn phages, despite their geographical distribution.

Therefore, there is a pressing need to determine which member of the order should be used as fecal human marker and if there are potential new candidates globally widespread. If this cosmopolitan group of phages are to be used for microbial source tracking and wastewater-based epidemiology[44], an accurate molecular marker should probably be better identified by the genetic characterization of isolated crAss-like phages and the definition of a core crAss-like phage genome. Otherwise, a large amount of data identified as crAssphage, but which actually belong to different crAss-like viruses, with no comparability between locations and no correlation with human fecal pollution and fecal microorganisms may be generated, with the risk that the potential of this new human-specific indicator remains unfulfilled.

The discovery of crAssphage in the human gut has been a showcase achievement of metaviromics[1,4]. By integrating culture-based and computational efforts, the isolation of new crAss-like phages allows research to go beyond metagenomics and the sequence variation hurdle and may provide new insights into the life cycle of these human-gut phages.

## Methods

No ethical approval was necessary for sample collection. The research presented complies with all relevant ethical regulations from the bioethics commission of the University of Barcelona.

### Strains, bacteriophages, media, and culture conditions

*Bacteroides* strains of different species (Supplementary Table 2) were used in propagation cultures as hosts to detect crAss-like phages from wastewater and to evaluate their sensitivity to crAssBcn phages. Exopolysaccharide-producing *B. intestinalis* strain APC919/174 (DSM 108646) was selected as the definitive host for the isolation of crAssBcn phages, and bacteriophage ΦCrAss001 (DSM 109066)[13] was used as the positive control.

Anaerobe basal broth (ABB) (Thermofisher) was used for the growth of the *Bacteroides* strains. For the plaque assays, ABB containing 0.7% (w/v) of agar-agar for soft agar or 1.4% (w/v) of agar-agar for the agar plates were used.

Manipulation was performed in aerobic conditions while incubation of cultures and agar plates was done at 37 °C for 24 or 48 h in anaerobic jars (GasPak; BBL) with $CO_2$ atmosphere generators (Anaerocult A; Merck).

### Wastewater samples

Twelve raw influent samples were obtained from five urban wastewater treatment plants (WWTPs) in Catalonia (NE Spain) over a period of one year. One of the WWTPs served a population of almost 384,000 inhabitants (Gavà); two plants served populations of between 100,000 and 290,000 inhabitants (Manresa and Igualada); and two plants served a population of more than 2,000,000 inhabitants (Besòs and Prat). Samples were collected in sterile containers, transported to the laboratory within two hours of collection and processed immediately for bacteriophage isolation as described below. Ten ml samples were filtered through 0.22 μm pore size, low-protein-binding (PES) membranes (Millipore) to remove bacteria and other particulate material.

### Bacteriophage enumeration

One ml of phage suspension from wastewater was used to infect one ml of cultures of *Bacteroides* cultures at the middle-exponential growth phase ($OD_{600}$ of 0.3) in 8 ml of ABB medium and incubated anaerobically and statically at 37 °C for 24 h. For subsequent enrichment cultures, 1 ml of the phage suspension from the first culture was filtered and used to infect a new *Bacteroides* culture under the same conditions.

For plaque assays, ten-fold serial dilutions of each phage suspension prepared with SM buffer (200 mM NaCl, 10 mM $MgSO_4$, 50 mM Tris-HCl, pH 7.5) were enumerated by the double agar layer method[45] with the respective *Bacteroides* host strains in ABB soft agar. Plates were incubated anaerobically at 37 °C for 24 h. Spot assays were performed as described for plaque assays but without the addition of

phage to the ABB soft agar. A 10 μl drop of phage suspension was directly applied to the solidified lawn of each host strain and dried prior to incubation. Negative controls were prepared without the addition of phage.

## Plaque blot hybridization

CrAss1-ORF46 probe is a digoxigenin (DIG)-labeled 50-bp probe (5′-ACCTGCTTCTACACTTTCCTTAGATGAACTAA-TATCTAACCCAGCTCTAT-3′) located in the ΦCrAss001 genome and commercially available. Plaques observed in the soft agar layer were transferred to a nylon membrane (Hybond N + , Amersham Pharmacia Biotech) and hybridized with the probe. Hybridization was performed at 53 °C, according to the standard procedure[46]. Stringent hybridization was achieved with the DIG-DNA Labeling and Detection Kit (Roche Diagnostics) according to the manufacturer's instructions.

Plaques showing a positive signal were recovered from the soft agar overlayer with a sterile loop, resuspended in 200 μl of SM buffer and submitted to a chloroform treatment to eliminate bacterial cells. The isolated plaques confirmed to be crAss-like phages by qPCR were further propagated in larger volumes of *B. intestinalis* culture for subsequent analysis.

## Purification of phage particles

Suspensions containing each individual phage were further purified by cesium chloride (CsCl) density gradients using ultra clear thin wall tubes (Beckman), 1 ml of 20% (w/v) sucrose and three densities of CsCl (1.3, 1.5, and 1.7 g/ml)[46]. Samples were ultracentrifuged at 22,000 x *g* for 2 h at 4 °C in a Swinging-Bucket SW-41 Rotor in a Beckman ultracentrifuge.

The visible grey bands corresponding to bacteriophages[46] were collected by puncturing the tube, obtaining a 0.5 ml volume that was dialyzed using prepared dialysis membranes (MWC 12–14 kDa) (Thermofisher) in dialysis buffer (Tris 0.1 M, EDTA 0.2 mM, pH 8) for 2 h. The dialysis buffer was replaced with fresh buffer and further dialyzed for 18 h with magnetic stirring.

## Infectivity assays

To evaluate the dynamics of infection of the 25 crAssBcn phages, each phage suspension was used to infect 1 ml of a middle-exponential growth phase culture of *B. intestinalis* grown in ABB at a multiplicity of infection (MOI) of 0.001. Tubes were incubated anaerobically at 37 °C, and the growth was monitored by measuring the OD of the cultures at 600 nm. In parallel, the number of colony-forming units grown in ABB agar, and the number of plaques of lysis plaques obtained by the double agar layer method (pfu/ml) were evaluated at intervals for 24 h. As a control, a *B. intestinalis* culture without phage was grown under the same conditions. All phage counts were normalized by the increase in concentration at each time point with respect to the initial value. Data were analyzed using GraphPad Prism 9 (GraphPad Software, www.graphpad.com). Comparisons between the average increase in the concentration of all crAssBcn phages *vs* ΦCrAss001 and among crAssBcn phages were evaluated with a Wilcoxon matched-pairs test and Friedman test with Dunn's multiple comparison test.

To calculate the burst size, adsorption assays were performed after the first burst (approximately at 150 min) at a MOI of 0.001 with *B. intestinalis* grown anaerobically to middle-exponential growth phase, in accordance with Kropinski[47], with modifications to adjust to the anaerobic conditions and the growth rate of *B. intestinalis*. Free phages were quantified by the double agar layer method as described above.

## Electron microscopy observations

Ten μl of the concentrated CsCl phage suspensions were dropped onto copper grids with carbon-coated Formvar films and negatively stained with 2% ammonium molybdate (pH 6.8) for 2 min. Phages were visualized using a Jeol 1010 TEM (JEOL Inc.) operating at 80 kV.

## Phage DNA isolation

DNA isolation was performed with the QIAamp DNA blood mini kit (Qiagen GmbH,), following the manufacturer's instructions. The DNA was suspended in a final volume of 200 μl of sterile bidistilled water. The DNA concentration of each pooled sample was evaluated using a Qubit® Fluorometer (Life Technologies) and the DNA quality was further confirmed by the 2100 Bioanalyzer system (Agilent Technologies).

## PCR and qPCR assays

A PCR assay (UP: 5′-ACCTGCTTCTACACTTTCCTT-3′/LP: 5′-AGTGCTCCAGAATAGGATTGT-3′) and a qPCR assay using TaqMan hydrolysis probe (UP: 5′-ACCTGCTTCTACACTTTCCTT-3′/LP: 5′-AGTGCTCCAGAATAGGATTGT-3′/Probe: 6FAM- ATATCTAACC-CAGCTC-MGBNFQ) was designed from the ΦCrAss001 genome (NC_049977.1) targeting a gene coding for a hypothetical protein (ORF 46) and previously described qPCR assays for the detection of crAss-like phages[12,19] were used. Primers and probes were confirmed as specific for the detection crAss-like phages available in genomic databases.

PCR amplification was performed using DreamTaq Green DNA Polymerase (Fermentas) in a GeneAmp PCR system 2700 (Applied Biosystems). qPCR amplifications were carried out using the standard run in the StepOne™ Real Time PCR System (Applied Biosystems) in a 20 μl reaction mixture with TaqMan® Environmental Master Mix 2.0 (Applied Biosystems). The reaction contained 9 μl of the sample DNA or standards with known DNA concentration prepared from gBlocks™ Gene Fragments used for quantification. The results were analyzed with the Applied Biosystems StepOne™ Instrument program. All samples were run in triplicate (including the standards and negative controls). The number of gene copies (GC) was defined as the mean of the triplicate data obtained.

## Sequencing

Five μl of DNA at a concentration of 0.2 ng/μl was fragmented and used to prepare libraries for whole genome sequencing with the Kapa Hyper Plus Kit (Roche) according to the manufacturer's protocol. Libraries were purified using AmPure beads (Beckman Coulter Inc.), checked for fragment distribution and size and quantified in a TapeStation 4200 and the Agilent High Sensitivity D1000 ScreenTape system (Agilent Technologies) in a Quantus™ Fluorometer (Promega). An equimolar pool of the individual 25 phage genomes was separately sequenced by NextSeq System (Illumina)with a high output run of 300 cycles.

## Sequence trimming, genome recovery and functional annotation

Raw reads were trimmed by Trimmomatic (LEADING:3 TRAILING:3 SLIDINGWINDOW:4:15 MINLEN:36)[48]. The quality of trimmed reads was checked by FastQC[49]. Paired-end reads were joined using fq2fa from the idba package v1.1.3[50]. Additionally, pair-end filtered reads were assembled individually by SPAdes v3.13.0 (-k 21,33,55,77,99,127)[51]. In order to recover additional complete viral genomes from the assembly sequences, genomes were assembled applying the Trusted contigs strategy[52], available in SPADEs, that used the ΦCrAss001 genome[13] as a reference to provide a guidance during de novo assembly. ORF prediction was carried out using Prodigal[53]. Genomic maps were generated using Geneious Prime version 20231.1.

## Clustering and intragenome comparison of crAssBcn phages

Phage genomes were clustered at 90% by cd-hit-est[54], the phage genome similarities were calculated using VIRIDIC[55] (BLASTn-based). VIRIDIC[55] calculates the intergenomic similarity of two viruses using BLASTn and checks each comparison for genomic synteny[56]. The reciprocal best match strategy was chosen for the ORF comparison in aminoacids, (rbm.rb script from enveomics package[57], selected default parameters). Taxonomic assignation of recovered phage genomes was

done by VIPtree (genome-wide similarity-based)[58]. Multiple alignments of selected protein sequences were performed by MUSCLE[59]. Phylogenetic tree reconstruction was drawn using IQ-TREE[60] and Geneious Prime 2023.1.1 (tree build method neighbor-joining). The polymerase phylogenetic analysis includes a database of polymerases A and B previously described[7]. Additionally, non-redundant database (updated in April 2023)[61] (BLASTp) best hit matches were also included in the analyses, considering those hits showing a genome coverage larger than 50% and an identity over 50%. In case that no sequences accomplished these criteria, then the ten best hits were selected.

## Functional protein annotation and structure prediction

Protein sequences were searched using the InterProScan software v5.47-82.0[62] to identify signatures from the InterPro member databases; Pfam[63], SMART[64], TIGRFAMs[65] and CDD[61]. Additionally, all crAssBcn phages ORFs were mapped against all ΦCrAss001 genes (annotation update January 2023) and taking in consideration recent identification of previously mis-annotated proteins encoded by the ΦCrAss001 genome[14] in order to obtain a more accurate functional gene annotation. Prediction of the structure and conserved active domains of the polymerase encoded in the crAssBcn phages was performed using Alphafold[66], and the NCBI's Conserved Domain Database and SPARCLE[67].

## Presence and abundance of crAssBcn phages around the world

To evaluate the geographical distribution of the crAssBcn phages, we performed a fragment recruitment of the crAssBcn phages against a viral metagenome collection that we created consisting of 1,255 human gut viromes from children and adults and in 14 countries (Supplementary Data 3). These metagenomes were mapped against 1) the genomes of the 25 crAssBcn phages pooled together (Pool), 2) the ΦCrAssBcn6 genome representing species 1, which formed the larger group of crAssBcn phages, and 3) the ΦCrAss001 genome, using standalone BLASTn with a cutoff of 70% query coverage, e-value ≥ 10⁻¹ and filtered by the 'best hit' option. Next, crAssBcn phage abundances were calculated using sequencing depth values normalized by dataset size (Gbp) and genome length (Kbp) (sequencing depth/Gbp/Kbp). Fragment recruitment data were plotted by the enveomics.R package in the R statistical tool[57]. Additionally, the graphs (bars, boxplots and heatmaps) were drawn with Plotly by R[68] and heatmapper[69].

## Statistics and Reproducibility

Statistical analyses were performed using the GraphPad Prism 9 (GraphPad Software, San Diego, CA, US). Unpaired t-tests were conducted to identify differences between treatments. Significant differences were set at $p < 0.05$. Experimental data presented are the average of three independent experiments. Bioinformatic analysis were performed in duplicate and are reproducible. At least five electron micrographs were taken for each phage, and a selected one for each phage is presented in Fig. 1.

No statistical method was used to predetermine sample size, no data were excluded from the analyses; the experiments were not randomized, and the Investigators were not blinded to allocation during experiments and outcome assessment.

## Reporting summary

Further information on research design is available in the Nature Portfolio Reporting Summary linked to this article.

## Data availability

The crAssBcn phages genomes generated in this study have been deposited in GenBank" with the GenBank accession codes available in Supplementary Table 3 and are publicly available. Other supporting data generated in this study (tree gene sequences, tree alignments and values) can be found in https://data.cyverse.org/dav-anon/iplant/ home/lolesramosub/Ramos-Barbero%2C%20MD%2C%20G%C3%B3mez-G%C3%B3mez%2C%2C.%2C%20Sala%2C%20L%20%28...%29%26%20Muniesa%2C%20M.%202023%20Supporting%20information/Ramos-Barbero%202023%20Supporting%20information.rar. The following databases were used: Pfam (http://pfam.xfam.org/), SMART (http://smart.embl-heidelberg.de/) TIGR-FAMs (https://www.jcvi.org/research/tigrfams) and CDD (https://www.ncbi.nlm.nih.gov/Structure/cdd/cdd.shtml). Database of metagenomes in Fig. 6 is available in Supplementary Data 3. Source data for supplementary figure 2 are provided in the Source Data file. Source data for supplementary figure 6 are provided in Supplementary Data 2. The authors declare that all other data supporting the findings of this study are available within the paper and its supplementary files. Source data are provided with this paper.

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

## Acknowledgements

This research was funded by the Spanish Ministerio de Innovación y Ciencia (PID2020-113355GB-I00) the Agencia Estatal de Investigación (AEI) and the European regional fund (ERF). C. G-G has a fellowship from the University of Barcelona. MD.R.B has a Margarita Salas fellowship from the Universitat d' Alacant. L. S-C has a Maria Zambrano fellowship. G. V has a FPI grant from the Spanish Ministerio de Universidades. S. M-C has a grant from Colciencias (Republic of Colombia) and L. R-R. is lecturer of the Serra-Hunter program, Generalitat de Catalunya. M. M. is a researcher of the ICREA Academia 2022 program. Authors want to acknowledge Prof. J. Anton and Dr. M. Martinez-Garcia from University of Alicante for their valuable help and support with the use of their servers for the bioinformatic analysis of this study, and Prof. F. Navarro and Prof. L. Comstock for some of the strains used in this study.

## Author contributions

M.D.R-B. conducted all bioinformatics analyses, prepared figures, and contributed to writing and revising the paper. C.G-G. isolated all phages, performed characterization and infectivity experiments and prepared figures. L.S.-C. conducted infection experiments, calculated all statistical analysis, prepared figures, and contributed to writing the paper. M.D-.R-B., C.G-G., and L.S-C. contributed equally to this work. L.R-R. wrote and revised the paper and discussed results. S.M-C. evaluated phages by qPCR and wrote sections of the paper. E.dM. and D.T-A. contributed to microscopy studies and phage characterization. G.V. performed the sampling, prepared media and performed some infection assays. A.R.B. gave technical support and conceptual advice and revised the paper. E.B. gave technical support and conceptual advice and commented on the manuscript at all stages. C.G-A. funded the study, discussed results, and commented on the manuscript at all stages. M.M. is the corresponding author, supervised and funded the study, designed experiments, analyzed data, and wrote the paper.

## Competing interests

The authors declare no competing interests.
