## [Peer Review File · Nature Communications]

REVIEWER COMMENTS

Reviewer #1 (Remarks to the Author):

A manuscript by Gomez-Gomez et al reports on isolation of a panel of 25 novel crAss-like phages closely related with crAss001 (*Kehishuvirus primarius*, family Steigviridae, order Crassvirales), their biological characterisation, and comparative analysis of their genomes. Unfortunately, the authors appear to be misguided about the currently adopted taxonomy of Crassvirales and use many terms incorrectly.

There should be a distinction made between crAssphage (p-crAssphage) as an individual phage strain (or a group of strains), belonging to the species *Carjivirus communis*, on one hand, and Crassvirales (crAss-like phages) as a diverse taxonomic group including dozens of different species, on the other hand. Not all of them are highly abundant in the human gut, or specific to the human species. p-crAssphage and its immediate relatives have never been isolated in culture. Most of the biological studies concerned with Crassvirales are performed on a handful of phage isolates (crAss001, DAC15 etc), which not only belong to a species, genus and even a family different from *Carjivirus communis*, but also are not as abundant in the human gut as p-crAssphage. Strains of bacteriophages isolated in this study are related with the genus *Kehishuvirus*, which makes them different from crAssphage sensu stricto (p-crAssphage, *Carjivirus communis*). Therefore, the use of of term "crAssphage" throughout the manuscript is very misleading and should be revised.

Authors approach to isolation of novel Crassvirales through enrichment of sewage samples on a large panel of *Bacteroides* strains was correct in principle. However, due to the use of a very narrowly specific screening procedure (PCR primers and a hybridisation probe specific to crAss001 gene 46), the authors missed a unique opportunity to discover and isolate a truly diverse collection of Crassvirales (perhaps p-crAssphage and other uncultured species), and instead focused on close relatives of an already reported species (*Kehishuvirus primarius*).

Nevertheless, I believe the some of the findings in this study are very interesting. In particular, I found it very intriguing to observe high intraspecies variability of certain specific genes and gene clusters, indicating rapid phage-host co-evolution.

Having said that, I believe that additional experiments, data analyses and a major revision of the manuscript are needed, before this manuscript can be considered for publication. The results of this study may be interesting to a more narrow audience of scientists working in the areas of phage biology, comparative genomics and systematics, as well as to microbiome scientists.

Major concerns:

- 1) The authors need to familiarise themselves with diversity of crAss-like phages, their taxonomy and principles of classification: <https://doi.org/10.1016/j.tim.2020.01.010> , <https://doi.org/10.1038/s41467-021-21350-w> , as well as <https://ictv.global/ictv/proposals/2021.022B.R.Crassvirales.zip> . That should help them to improve Discussion.
- 2) Re-doing or perhaps correctly reporting one-step growth curves data is highly recommended.
- 3) Comparing the genomes of 25 isolates with currently adopted (and largely uncultured) 12 genera and 15 species in subfamily Asinivirinae of the family Steigviridae includes (<https://ictv.global/ictv/proposals/2021.022B.R.Crassvirales.zip>). This could be one of the major results in this study.

Specific comments:

Line 24. change "crAssphage is an abundant..." to "Crassvirales (crAss-like phages) are an abundant group of..."

Line 25: "crAss-like phages" instead of "crAssphage"

Line 26: "four cultured strains of crAss-like phages" instead "four crAssphage virions"

Line 27 and throughout the manuscript. The term virion denotes an individual phage particle. Please change "virion" to "isolate" or "strain" in similar context throughout the manuscript.

Line 30: There's no such thing as "lytic plaques". There is however a "lytic cycle" of a virulent or a temperate phage. Just use "plaques" instead.

Line 32: "crAss-like phages" instead of "crAssphages"

Line 45: "crAss-like phages" instead of "crAssphage"

Line 48: "isolating crAss-like phages in pure culture" instead of "isolating crAssphage virions"

Line 50-51: "[isolated] on Bacteroides..." instead of "[isolated] from Bacteroides..."

Line 51. Two additional isolates of crAss-like phages were recently reported by Hedzetz et al (<https://doi.org/10.1038/s41598-022-25636-x>)

Line 56: "five subfamilies with ten genera..." This is no longer correct. Please refer to the ICTV taxonomy for currently accepted taxonomic structure of Crassvirales

Lines 59-60: Change "Plaques of Φ CrAss001 are only visible after propagation in liquid medium, but the infected bacterial cultures are not cleared" to "Plaques of Φ CrAss001 are visible in agar overlays, but the infected liquid bacterial cultures are not cleared".

Line 69: Change "crAssphage-like sequences" to "crAss-like phage genome sequences"

Line 70: Change "crAssphages" to "crAss-like phages, especially p-crAssphage"

Line 71: Change "crAssphage" to "p-crAssphage".

Line 84: Change "virulent crAssphage virions" to "virulent crAss-like phages"

Line 88: Change "Isolation of crAssphage" to "Isolation of crAss-like phages"

Line 89: "crAss-like phages" instead of "crAssphage"

Line 90: It is unclear what set of primers was used here. What were they targeting? The only set of primers described in the M&M section seems to be targeting crAss001 and not p-crAssphage or other Crassvirales.

Line 92: What molecular analyses were used here?

Line 96: If the pair of primers mentioned in my comment above, along with a hybridisation probe specific to crAss001 gene 46, were used for this screening, no wonder that the only group of phages detected in the enrichment were phages closely related with crAss001. If other Crassvirales were enriched in culture of Bacteroides of different species (remaining 70-80% of the plaques), they would have been automatically overlooked. Screening by sequencing of VLP DNA could be a much more productive approach for obtaining a read out in this screening. I think this was a big mistake by the author to not look for other Crassvirales.

Line 114: Family Podoviridae has been abolished in the latest revision of ICTV taxonomy. Please replace with "podovirus-like morphology".

Lines 117-120: One-step growth curves are said to be available, but not presented here. Or is it the Fig. 2A that the authors refer to as one-step growth curves. In that case these are not one-step growth curves. One-step growth curve is meant to be an observation of a single generation period, with closely spaced time points. The data presented in Fig. 2A gives no information on the phage latency period or burst size.

The scale for [log₁₀] cfu/ml in Fig. 2B should be changed, so that the minimum value is 8 not 1.

Line 135: "was only slightly higher" - This, in fact, might be a dramatic difference considering that the inoculum titre was >10⁸ cfu/ml.

Line 139: "22 complete crAss-like phage genomes" instead of "22 complete crAssphage genomes"

Line 144: Difference in nucleotide sequence does not necessarily (!) lead to difference in aa sequence, due to codon degeneracy. Please rephrase this statement.

Lines 147-149: "order Caudovirales, family Podoviridae, subfamily Betacrassvirinae and genus VI" - this is a blend of two outdated taxonomies. Please remove.

Line 152: Subfamily Asinivirinae in the family Steigviridae currently includes 12 genera and 15 species (<https://ictv.global/ictv/proposals/2021.022B.R.Crassvirales.zip>). It would be very helpful if the authors could compare their 25 isolates with type genomes assigned within those 15 species, to

check if there is any overlap between them. Assigning new cultured type strains to some (or all) of these species could be a major result of this study.

Line 180-185 and Figs. 4 and 5: I strongly recommend the authors to check this recent pre-print of a study that used cryoEM microscopy to identify many of the previously mis-annotated proteins encoded by the crAss001 genome (<https://www.researchsquare.com/article/rs-1898492/v1>).

Line 208: "crAss-like phages" instead of "crAssphages".

Lines 194-195: "This lack of homology raises questions about the criteria used to classify crAssphage, which is discussed later." - The lack of nucleotide sequence homology between different clades of Crassvirales not surprising given the high evolution rates in phages, and therefore should not raise questions.

Line 218: Here and throughout the manuscript, the authors seem to miss the difference between p-crAssphage, and Crassvirales, including crAss001. I would recommend the authors to get acquainted with these works in order to improve their understanding of Crassvirales diversity and taxonomy: <https://doi.org/10.1016/j.tim.2020.01.010> , <https://doi.org/10.1038/s41467-021-21350-w> , as well as <https://ictv.global/ictv/proposals/2021.022B.R.Crassvirales.zip>

Line 221-222: To further stress my point about taxonomic confusion above - Φ CrAss001 is not a crAssphage!

Line 221-225: Φ CrAss001 infects *B. intestinalis*. The authors chose to use a very narrowly specific pair of primers and hybridisation probe that allowed them to detect and isolate only phages closely related to crAss001. Unsurprisingly, all of them were infecting *B. intestinalis*!

Lines 223-336. Replace "crAssphage" with "crAss-like phages" everywhere. crAssphage was not an object of this study...

Line 326: "Therefore, there is a pressing need to establish a minimum level of genomic coverage and homology for crAssphage classification." - Due to high rate of evolution, nucleotide sequences of phages belonging to the same family or order diverge beyond recognition. It is impossible therefore to classify phages at the level of order or family (or even genus in many cases) using a series of nucleotide sequence identity cut-offs. Protein sequence phylogenies and gene sharing networks are typically applied instead. Please consult with <https://doi.org/10.1016/j.tim.2020.01.010> , <https://doi.org/10.1038/s41467-021-21350-w>

Reviewer #2 (Remarks to the Author):

The manuscript reports the isolation, sequencing and geographical distribution of 25 crAssBcn phages. These phages are very similar (80-91% nucleotide identity) to previously reported isolate crAss001 – which was expected since the isolates were selected with plaque spot hybridization using

a CrAss001 probe. Since crAss001 genome has been rigorously studied, genome annotation and 'intra-genomic comparison' of the new phages have little merit per se. What would be interesting to see is the study of the differences between these closely related and overall syntenic genomes. What proteins are present in some but not all crAssBcn genomes (Figure S3)? What are the high variability proteins (<50%AAI, Figure S4)? The authors have mentioned DNA polymerase being among these highly variable genes. This might be a valuable observation, if followed with sequence analysis of DNA polymerases of this phage groups (e. g., multiple sequence alignment with functional motifs marked).

"Presence and abundance of crAssBcn phages around the world" was calculated by mapping metagenomes on crAssBcn and crAss001 genomes; it does not cover the whole diversity of crAss-like phages present in human gut metagenomes. Hence, "The widespread prevalence of crAssBcn phages" should not be claimed.

Tables and figures.

Figure S2, "Phylogenetic relationship of the CrAssBcn phages" might be made a main figure, because the reader would like to see where these new isolates stand compared to previous ones.

Table 1 contains technical details on CrAssBcn phage isolation and could be moved to Supplement.

Figure 1. "Morphology of the new crAssBcn phages". Since the CrAssBcn genomes are very similar to the original CrAss001, Podoviridae morphology was anticipated; Figure 1 containing 25 pictures of the crAssBcn virions could be omitted or moved to Supplement.

Figure 2. "Infectivity dynamics of the new crAssBcn phages". The infectivity of the new phages, by authors' words, "did not differ significantly from that of Φ CrAss001". This figure should be moved to Supplement.

Figure 4. "Circular genomic map of Φ CrAssBcn6". This figure is not needed, since this genome, as all crAssBcn, is very similar to Φ CrAss001. A figure depicting the differences between these genomes (and corresponding discussion) would be more appropriate.

Figure 5. "Comparison of the ORFs (reciprocal best match) of the seven crAssBcn species." the figure is not informative and should be omitted. Again, a study of the differences between these closely related genomes is anticipated.

Minor comments.

Lines 29-31. "The crAssBcn phages generate turbid lytic plaques and show similar propagation, reaching titers of 10⁹ pfu/ml <...>" – should be moved from Abstract to Results section.

Lines 31-32. "CrAssBcn phage genomes are similar to Φ CrAss001 but differ from those of other crAssphages." – specify in what respect they are similar or different.

Line 143. “Functional annotation of all phages by Blastp is presented in Table S3.” Blastp should not be used for functional annotation, since blastp hits might be misannotated. Instead, the curated annotation of Φ CrAss001 might be used.

Lines 191-195. “Comparison with 64 non-redundant crAss-like phage genomes in the databases (Fig. S5) showed clear differences. Moreover, relevant differences were also observed between the phage genomes identified as crAssphages in the databases (Fig.S5), with some shared identities restricted to only one or a few ORFs. This lack of homology raises questions about the criteria used to classify crAssphage, which is discussed later.” – Protein sequence similarity and homology are not the same; establishing homology requires more elaborate steps than creating an all-vs-all similarity matrices. Since the crAssBcn genomes are very similar to CrAss001, and several studies on comparative genomics of crAss-like phages including CrAss001 have already been published, this section should be omitted; the “lack of homology” claim is incorrect.

Lines 208-228 belong to Introduction, not Discussion.

Lines 281-283. “Another unexpected result was the difference in the AAI (< 50 %) of the DNA polymerase I gene of species V and VI compared to the other crAssBcn phages, considering they all belong to the same subfamily and infect the same host.”

Line 323. “This brings us to the absence of criteria to define what a crAssphage is.” Again, there are several studies dedicated to this issue.

Lines 469-471. “In order to recover additional complete viral genomes from the assembly sequences, genomes 20 and 25 were assembled applying the ‘-trusted’ option and employing the Φ CrAss001 genome as a reference.” Please provide the reasoning.

Lines 472-474. “The genome completeness was checked by checkcomplete and VIBRANT. ORF prediction was carried out using Prodigal and functional annotation of predicted genes was performed by Diamond Blastp using the NR NCBI database” Since crAss-like phages have circular genomes, the genome completeness is straightforward: the assembly should be circular. Again, blastp should not be used for functional annotation.

In particular, we -the editors- expect the revisions to provide the requested additional analyses, adjust the text and claims with regards to the nomenclature, and to tone down the claims on the phages being widespread.

A/We thank the editors for the opportunity of revising our manuscripts. All requirements of the Reviewers have been considered and addressed and are indicated below.

REVIEWER COMMENTS

Reviewer #1 (Remarks to the Author):

A manuscript by Gomez-Gomez et al reports on isolation of a panel of 25 novel crAss-like phages closely related with crAss001 (Kehishuvirus primarius, family Steigviridae, order Crassvirales), their biological characterisation, and comparative analysis of their genomes. Unfortunately, the authors appear to be misguided about the currently adopted taxonomy of Crassvirales and use many terms incorrectly.

There should be a distinction made between crAssphage (p-crAssphage) as an individual phage strain (or a group of strains), belonging to the species Carjivirus communis, on one hand, and Crassvirales (crAss-like phages) as a diverse taxonomic group including dozens of different species, on the other hand. Not all of them are highly abundant in the human gut, or specific to the human species. p-crAssphage and its immediate relatives have never been isolated in culture. Most of the biological studies concerned with Crassvirales are performed on a handful of phage isolates (crAss001, DAC15 etc), which not only belong to a species, genus and even a family different from Carjivirus communis, but also are not as abundant in the human gut as p-crAssphage. Strains of bacteriophages isolated in this study are related with the genus Kehishuvirus, which makes them different from crAssphage sensu stricto (p-crAssphage, Carjivirus communis). Therefore, the use of term "crAssphage" throughout the manuscript is very misleading and should be revised.

A/The reviewer has obviously more experience than us in Crassvirales taxonomy and we apologize for not adhering to the new taxonomy. Although we revised it just before submitting the manuscript, we were not aware of the different designation between crAss-like phages and p-crAssphage. We have now revised it throughout in the manuscript and the reviewers's comments have been taken into consideration and corrected accordingly.

Authors approach to isolation of novel Crassvirales through enrichment of sewage samples on a large panel of Bacteroides strains was correct in principle. However, due to the use of a very narrowly specific screening procedure (PCR primers and a hybridisation probe specific to crAss001 gene 46), the authors missed a unique opportunity to discover and isolate a truly diverse collection of Crassvirales (perhaps p-crAssphage and other uncultured species), and instead focused on close relatives of an already reported species (Kehishuvirus primarius).

A/Please, see comments below where we explain this point more extensively. In addition, we have added some extra information in the paper.

Nevertheless, I believe the some of the findings in this study are very interesting. In particular, I found it very intriguing to observe high intraspecies variability of certain specific genes and gene clusters, indicating rapid phage-host co-evolution.

Having said that, I believe that additional experiments, data analyses and a major revision of the manuscript are needed, before this manuscript can be considered for publication. The results of this study may be interesting to a more narrow audience of scientists working in the areas of phage biology, comparative genomics and systematics, as well as to microbiome scientists.

A/We thank the reviewer for the positive comments. Additional experiments, completely new figures, new analysis and a major revision of the manuscript have been performed now. We hope the reviewer will agree.

Major concerns:

1) The authors need to familiarise themselves with diversity of crAss-like phages, their taxonomy and principles of classification: <https://doi.org/10.1016/j.tim.2020.01.010> , <https://doi.org/10.1038/s41467-021-21350-w> , as well as <https://ictv.global/ictv/proposals/2021.022B.R.Crassvirales.zip> . That should help them to improve Discussion.

A/We agree with the reviewer and our revised manuscript has been updated with all the required information. The discussion has been extensively modified accordingly.

2) Re-doing or perhaps correctly reporting one-step growth curves data is highly recommended.

A/ Our mistake. As we confirm below, the reviewer refers to the charts in Figure 2 and is completely right. These are not one-step growth curves, but phage propagation dynamics of the phages for 24 hours. Nevertheless, charts in Figure 2 have been moved to supplemental material by request of another reviewer.

As indicated below, burst size of the phages was calculated according to Kropinski. As suggested by the reviewer, we have performed new replicates of our experiments and by increasing the replicates we have obtained a narrower burst size range, now 64.6 ± 16 . This has been correctly indicated in the revised manuscript.

3) Comparing the genomes of 25 isolates with currently adopted (and largely uncultured) 12 genera and 15 species in subfamily Asinivirinae of the family Steigviridae includes (<https://ictv.global/ictv/proposals/2021.022B.R.Crassvirales.zip>). This could be one of the major results in this study.

A/We thank the reviewer for the suggestion. We have now compared our genomes with the 15 species of *Asinivirinae* subfamily as recommended (we thank also for providing the table with the necessary data). Actually, our previous comparison with genomes of crAss-like phages already included most of these genomes.

A new figure (Figure 3) has been built where the 25 crAssBcn and the 15 species in *Steigviridae* family have been used as query. The proteomic tree, visualized by VipTREE confirms that the 25 crAssBcn phages does not match with any of the previous species defined and only shows crAss001 in the same branch. In a new figure (Figure 4) we have aligned the genomes of the 15 members of *Steigviridae* family indicated with one of our phages Φ CrassBcn6, and it allows the visualization of the differences with the phages assigned to *Steigviridae* species. We hope the reviewer will agree.

Specific comments:

Line 24. change "crAssphage is an abundant..." to "Crassvirales (crAss-like phages) are an abundant group of..."

A/It has been changed

Line 25: "crAss-like phages" instead of "crAssphage"

A/ It has been replaced

Line 26: "four cultured strains of crAss-like phages" instead "four crAssphage virions"

A/Done

Line 27 and throughout the manuscript. The term virion denotes an individual phage particle. Please change "virion" to "isolate" or "strain" in imilar context throughout the manuscript.

A/It has been removed here and throughout the manuscript.

Line 30: There's no such thing as "lytic plaques". There is however a "lytic cycle" of a virulent or a temperate phage. Just use "plaques" instead.

A/ The referee is right, it has been corrected

Line 32: "crAss-like phages" instead of "crAssphages"

A/Done

Line 45: "crAss-like phages" instead of "crAssphage"

A/Done

Line 48: "isolating crAss-like phages in pure culture" instead of "isolating crAssphage virions"

A/Done

Line 50-51: "[isolated] on Bacteroides..." instead of "[isolated] from Bacteroides...".

A/It has been corrected

Line 51. Two additional isolates of crAss-like phages were recently reported by Hedzet et al (<https://doi.org/10.1038/s41598-022-25636-x>)

A/Originally, we have not included this reference since the study does not actually report the isolation of two culturable crAss-like phages in pure culture, but they detected two complete and two partial crAss-like genomes when analyzing the metagenome of the supernatant of an enrichment culture done on *Bacteroides uniformis*, which is not the same *in senso stricto* than having isolated individual new phages in pure cultures. What they report is a mixed phage suspension.

In contrast, during the review process a new pre-print appeared (6th March) with the description of three new isolated crAss-like phages (that are indicated as CrAssphages in the pre-print), infecting *Bacteroides cellulosilyticus*, that we have included in our reference list despite it is not yet published.

Line 56: "five subfamilies with ten genera..." This is no longer correct. Please refer to the ICTV taxonomy for currently accepted taxonomic structure of CrAssvirales

A/We apologize again for the lack of accuracy. It has been corrected and referenced.

Lines 59-60: Change "Plaques of Φ CrAss001 are only visible after propagation in liquid medium, but the infected bacterial cultures are not cleared" to "Plaques of Φ CrAss001 are visible in agar overlays, but the infected liquid bacterial cultures are not cleared".

A/Right, it has been changed

Line 69: Change "crAssphage-like sequences" to "crAss-like phage genome sequences"

A/Done

Line 70: Change "crAssphages" to "crAss-like phages, especially p-crAssphage"

A/Done

Line 71: Change "crAssphage" to "p-crAssphage".

A/Done

Line 84: Change "virulent crAssphage virions" to "virulent crAss-like phages"

A/Done

Line 88: Change "Isolation of crAssphage" to "Isolation of crAss-like phages"

A/Done

Line 89: "crAss-like phages" instead of "crAssphage"

A/Done

Line 90: It is unclear what set of primers was used here. What were they targeting? The only set of primers described in the M&M section seems to be targeting crAss001 and not p-crAssphage or other Crassvirales.

A/It has been clarified in the text that three qPCR assays were used but that the qPCR designed from crAss001 was the one ultimately working for our purposes.

As explained later in the discussion, we chose to use a qPCR targeting crAss001 because it was the first model phage/host bacteria available, however it was not the only attempt we did, although it was not included in the original manuscript.

For the isolation of individual crAss-like phages the requirements were:

1) to propagate the phages in the right host strain and to see an increase in the number of phages after propagation. For this we chose the qPCR because of its easy and quick application that allowed daily monitoring of the phage propagation in the enrichment cultures.

2) to get a phage suspension containing a sufficiently predominant proportion of crAss-like phages among the total number of phages in the suspension. This is because the suspension has to be plated by double agar layer and, to obtain separated plaques, the suspension has to be diluted. If the proportion of crAss-like phages among all other phages propagating in this strain is too low, the crAss-like phages fall under the detection limit (no plaques are generated by crAss-like phages), and it is impossible to isolate them. For this purpose, we needed first to enrich selectively crAss-like phages and monitor their enrichment before plating it.

Although data was not presented in the original manuscript, in addition to the qPCR designed from crAssphage001 we performed numerous attempts to isolate crAss-like phages using other two qPCR assays from other crAssphages (those described in García-Aljaro et al., 2017 and in Stachler et al., 2018) and using different hosts strains from our collection. This information has been now included

The problem was that without knowing the host strain to propagate these phages, most of these attempts were unsuccessful (we did not see any increase). We observed some increase using other *Bacteroides* strains from our collection, but it was never enough to enrich the crAss-like phages to a sufficient proportion to allow their isolation from the agar layers. The main question, we believe is that too many other non-crAss-like phages infected these strains, leaving the crAss-like phages in too low proportion to allow their isolation.

The only situation in which we were able to propagate crAss-like phages selectively and obtain them at a sufficient proportion for their isolation was using *B. intestinalis* and targeting them with the qPCR assay of crAss001. The reason why, we do not know exactly. We believe that this strain should have a narrow range of phages infecting it, with a higher proportion of them being crAss-like phages, allowing us to specifically propagate these phages in higher proportions versus other non-crAss-like phages.

Of course, the reviewer is right, and the use of this qPCR assay in particular limits the range of phages isolated and biases towards the isolation of crAss001-like phages, but in our hands, this was the only option to accomplish our objectives. Considering the limited number of crAss-like isolates described in the literature, this should also be the case in other authors' hands.

We have included some explanations in the manuscript to show that we have attempted to use other qPCR assays and other hosts without success. And we discuss that the use of this combination was not random, but our only chance to get the isolation of new viruses.

Line 92: What molecular analyses were used here?

A/qPCR, it has been corrected.

Line 96: If the pair of primers mentioned in my comment above, along with a hybridisation probe specific to crAss001 gene 46, were used for this screening, no wonder that the only group of phages detected in the enrichment were phages closely related with crAss001. If other Crassvirales were enriched in culture of *Bacteroides* of different species (remaining 70-80% of the plaques), they would have been automatically overlooked. Screening by sequencing of VLP

DNA could be a much more productive approach for obtaining a read out in this screening. I think this was a big mistake by the author to not look for other Crassvirales.

A/As commented above, we agree with the reviewer that sequencing VLP from the supernatants would have provided more information about the phages enriched in each host, and actually we do not discard this approach in future studies. Actually, this is the strategy described in Hedzet et al (mentioned above by the reviewer, but it does not allow the isolation of crAss-like phages in pure culture, but to describe crAssphage genomes after an enrichment culture.

Therefore, sequencing of VLP DNA was not useful for phage isolation, that was our goal. Moreover, the use of this particular qPCR/host was not random, two other qPCR assay were used with all the strains in our collection. Their use was not a mistake, it was because we wanted to isolate and characterize new crass-like phages, not to describe new crAssphage sequences after propagation in cultures. Please see comments above.

The suggestion of the reviewer is very interesting and broadens the spectrum of the phages detected. However, the lack of individual phages to work limits the studies of the crass-like phage biology, that is one of our interests and apparently one of the limitations in this phage research, considering the limited number of isolates described. As mentioned in the manuscript, we believe that the combined culturable and genomic approach is the added value

Line 114: Family Podoviridae has been abolished in the latest revision of ICTV taxonomy. Please replace with "podovirus-like morphology".

A/The reviewer is right, it has been corrected

Lines 117-120: One-step growth curves are said to be available, but not presented here. Or is it the Fig. 2A that the authors refer to as one-step growth curves. In that case these are not one-step growth curves. One-step growth curve is meant to be an observation of a single generation period, with closely spaced time points. The data presented in Fig. 2A gives no information on the phage latency period or burst size.

A/We apologize, this point was not well indicated. Please see our answer above. We have performed more experimental replicates that allowed us to narrow the burst size calculations, despite the variability of results obtained.

The scale for [log₁₀] cfu/ml in Fig. 2B should be changed, so that the minimum value is 0 not 1.

A/Absolutely it has been corrected. Now it correspond to Supplementary Figure 2

Line 135: "was only slightly higher" - This, in fact, might be a dramatic difference considering that the inoculum titre was >10⁸ cfu/ml.

A/Sorry, in this point we cannot agree with the reviewer. When talking about bacterial populations and using log units, a change of 0.3 log units on average as we observed here, could not be considered as a dramatic difference. Particularly, if compared with many other phages that after propagation using the same MOI cause differences in the host population of more than 6 log₁₀ units compared with the uninfected control. A difference of 0.3 logs is very narrow indeed.

Line 139: "22 complete crAss-like phage genomes" instead of "22 complete crAssphage genomes"

A/It has been corrected

Line 144: Difference in nucleotide sequence does not necessarily (!) lead to difference in aa sequence, due to codon degeneracy. Please rephrase this statement

A/Right, it has been rephrased

Lines 147-149: "order Caudovirales, family Podoviridae, subfamily Betacrassvirinae and genus VI" - this is a blend of two outdated taxonomies. Please remove.

A/It has been removed and the paragraph rephrased

Line 152: Subfamily Asinivirinae in the family Steigviridae currently includes 12 genera and 15 species (<https://ictv.global/ictv/proposals/2021.022B.R.Crassvirales.zip>). It would be very helpful if the authors could compare their 25 isolates with type genomes assigned within those 15 species, to check if there is any overlap between them. Assigning new cultured type strains to some (or all) of these species could be a major result of this study.

A/We thank the reviewer for this valuable suggestion. Please see above our answer. We have now analyzed the 25 isolates compared with the 15 species within the subfamily. Results are shown in figures 3 and 4 where it can be observed that crAssBcn phages do not match with any other previously species, except crAssBcn20, as already reported, therefore, they can be considered as new ones.

Line 180-185 and Figs. 4 and 5: I strongly recommend the authors to check this recent pre-print of a study that used cryoEM microscopy to identify many of the previously mis-annotated proteins encoded by the crAss001 genome (<https://www.researchsquare.com/article/rs-1898492/v1>).

A/We thank the reviewer for the suggestion and the information. In Supplemental Database 1, supplemental Database 3 and throughout the manuscript where we describe the annotation of crAss001, the list of proteins has been updated according to 1) the new version of annotated crAss001 genome released in 2023 and 2) also with the newly identified proteins indicated in this study suggested by the reviewer.

Moreover, the previously mis-annotated proteins identified in the suggested study have been checked and two of them correspond to those in which we observed the lowest %AAI. Therefore, the new figure 5 incorporates this modification. Other proteins considered to be conserved in the different families (Fig 1D of the pre-print), particularly in *Steigviridae* family, are also conserved within crAssBcn phages.

The pre-print has been cited in the study.

Line 208: "crAss-like phages" instead of "crAssphages".

A/It has been corrected

Lines 194-195: "This lack of homology raises questions about the criteria used to classify crAssphage, which is discussed later." - The lack of nucleotide sequence homology between different clades of Crassvirales not surprising given the high evolution rates in phages, and therefore should not raise questions.

A/We agree with the point raised by the reviewer. We intended to indicate that in the present moment different members of the clade are indistinctly used and proposed as fecal human marker, and that this should be considered, since not all of them are the same virus. The sentence has been rephrased and the discussion in general modified. We hope the reviewer will agree.

Line 218: Here and throughout the manuscript, the authors seem to miss the difference between p-crAssphage, and Crassvirales, including crAss001. I would recommend the authors to get acquainted with these works in order to improve their understanding of Crassvirales diversity and taxonomy: <https://doi.org/10.1016/j.tim.2020.01.010> , <https://doi.org/10.1038/s41467-021-21350-w> , as well as <https://ictv.global/ictv/proposals/2021.022B.R.Crassvirales.zip>

A/We thank the reviewer for the suggestion, as we mentioned above, we have revised and updated the manuscript accordingly.

Line 221-222: To further stress my point about taxonomic confusion above - Φ CrAss001 is not a crAssphage!

A/Again, we apologize for our lack of rigor adopting the new classification. We have intended to correct all the manuscript following the suggestions of the reviewer and here we have rephrased the sentence.

Line 221-225: Φ CrAss001 infects *B. intestinalis*. The authors chose to use a very narrowly specific pair of primers and hybridisation probe that allowed them to detect and isolate only phages closely related to crAss001. Unsurprisingly, all of them were infecting *B. intestinalis*!

A/Please, see the comments above. Just to make it clear, it was not a random choice, we are aware that this would limit the outcome of the study, but it was the only strategy that allowed us to isolate individual new phages. We believe that the characteristics of *B. intestinalis*, with a narrower range of phages infecting it, allowed to enrich specifically crAss-like phages. Previous attempts using other hosts and other qPCR assays and probes were unsuccessful, as we have now indicated in the manuscript.

Lines 223-336. Replace "crAssphage" with "crAss-like phages" everywhere. crAssphage was not an object of this study...

A/We have replaced it throughout the manuscript. We encountered the problem that actually some genomes are defined as crAssphage in the databases, and also in several environmental studies they indicate they are targeting crAssphage as human fecal marker. In both cases because they were published before the new taxonomical classification and probably because many researchers are not aware of the variability of the clade we have specified it. We hope the reviewer will agree.

Line 326: "Therefore, there is a pressing need to establish a minimum level of genomic coverage and homology for crAssphage classification." - Due to high rate of evolution, nucleotide sequences of phages belonging to the same family or order diverge beyond recognition. It is impossible therefore to classify phages at the level of order or family (or even genus in many cases) using a series of nucleotide sequence identity cut-offs. Protein sequence phylogenies and gene sharing networks are typically applied instead. Please consult with <https://doi.org/10.1016/j.tim.2020.01.010> , <https://doi.org/10.1038/s41467-021-21350-w>

A/We completely agree with the reviewer. The classification shown in the previous manuscript was already based on protein sequence, not on nucleotide sequence (as indicated in the figures). But it was incorrectly described, and we have now clarified this point.

A/We want to thank the reviewer for the accurate revision of our study, the useful suggestions and taxonomical clarifications as well as all the new material provided so that we can improve our study. We hope we have answered satisfactorily to all the questions and requirements.

Reviewer #2 (Remarks to the Author):

The manuscript reports the isolation, sequencing and geographical distribution of 25 crAssBcn phages. These phages are very similar (80-91% nucleotide identity) to previously reported isolate crAss001 – which was expected since the isolates were selected with plaque spot hybridization using a CrAss001 probe. Since crAss001 genome has been rigorously studied, genome annotation and ‘intra-genomic comparison’ of the new phages have little merit per se.

What would be interesting to see is the study of the differences between these closely related and overall syntenic genomes. What proteins are present in some but not all crAssBcn genomes (Figure S3)?

A/These genes present in only some of the crAssBcn genomes are shown in Supplementary Data 2, and we have now prepared a new table (Supplementary Table S5) showing the different genes that are not present in any of the 25 crAssBcn phages and indicating in which phage are present or absent. We hope the reviewer would find this presentation more informative.

What are the high variability proteins (<60%AAI, Figure S4)? The authors have mentioned DNA polymerase being among these highly variable genes. This might be a valuable observation, if followed with sequence analysis of DNA polymerases of this phage groups (e. g., multiple sequence alignment with functional motifs marked). –

A/Although this data was presented in the former figure 5 and in the former Table S4), we understand the reviewer suggests that a better and clearer visualization of these results is needed. Therefore, we keep former table S4 (now supplementary Data 2) since the data is presented in detail. But we have identified those ORF showing the lowest %AAI, that are mentioned in the text and discussion. From these, we have selected six with the lowest %AAI and encoding different type of proteins, and we have prepared a new figure (Fig. 5) showing heatmaps comparing each gene of each species against the other six species. We believe this allows a much better comparison of the differences observed.

Moreover, specifically for the polymerase, we have prepared a new figure (Fig. 6), where the sequences of polymerases of species I-VI are compared and aligned, and a new supplementary figure (supplementary Fig. 5) showing the conserved functional motifs in the two types of polymerases.

Further discussion has also been included. We hope the reviewer agrees with all these changes.

“Presence and abundance of crAssBcn phages around the world” was calculated by mapping metagenomes on crAssBcn and crAss001 genomes; it does not cover the whole diversity of crAss-like phages present in human gut metagenomes. Hence, “The widespread prevalence of crAssBcn phages” should not be claimed.

A/Actually, we have not indicated that they are the most abundant or much more abundant than other crAss-like phages, since indeed, not all crAss-like phages were analyzed and compared. We have indicated that these phages are quite abundant, more than crAss001 and are found in many different metagenomes around the globe. Nevertheless, we have revised our claims and made the expressions less enthusiastic. We hope the reviewer will agree.

Tables and figures.

Figure S2, “Phylogenetic relationship of the CrAssBcn phages” might be made a main figure, because the reader would like to see where these new isolates stand compared to previous ones.

A/A new figure has been built (Figure 3) including now comparison of crAssBcn phages and in addition we have included 15 crAss-like phages of the same genus Kehishuvirus, as a query, by request of other reviewers.

Table 1 contains technical details on CrAssBcn phage isolation and could be moved to Supplement.

A/No problem, it has been moved to supplemental material (Supplemental Table 1)

Figure 1. “Morphology of the new crAssBcn phages”. Since the CrAssBcn genomes are very similar to the original CrAss001, Podoviridae morphology was anticipated; Figure 1 containing 25 pictures of the crAssBcn virions could be omitted or moved to Supplement.

A/Actually we have removed this figure and few electron micrographs have been selected to illustrate the morphology of our isolates in a new Figure 1 where also their genomes are aligned. We believe it is important to keep some pictures to illustrate that we have been working with viral isolates, not just with genomes. We hope the reviewer will agree.

Figure 2. “Infectivity dynamics of the new crAssBcn phages”. The infectivity of the new phages, by authors’ words, “did not differ significantly from that of Φ CrAss001” . This figure should be moved to Supplement.

A/It has been moved

Figure 4. “Circular genomic map of Φ CrAssBcn6”. This figure is not needed, since this genome, as all crAssBcn, is very similar to Φ CrAss001. A figure depicting the differences between these genomes (and corresponding discussion) would be more appropriate.

A/The figure has been moved to supplementary material, we believe it can help to understand some parts of the manuscript. We hope the reviewer will agree.

As requested, four new figures (Fig. 1, Fig 3, Fig 5 and Fig 6) comparing the genomes of crAssBcn phages, proteomic tree, comparison of their ORF and specifically comparing the polymerase have been built, highlighting their differences in more detail, and the results are discussed.

Figure 5. “Comparison of the ORFs (reciprocal best match) of the seven crAssBcn species.”the figure is not informative and should be omitted. Again, a study of the differences between these closely related genomes is anticipated.

A/The study was done and shown in the figure and in former supplemental table S4 (Excel file) (now supplementary data 2). However, we understand that the reviewer felt that this figure did not present the results with clearly enough. As indicated above, several figures have been constructed to study the differences between the phage genomes. We hope the reviewer agrees with the modifications and consider them more informative.

Minor comments.

Lines 29-31. “The crAssBcn phages generate turbid lytic plaques and show similar propagation, reaching titers of 109 pfu/ml <...>” – should be moved from Abstract to Results section.

A/ It has been moved

Lines 31-32. “CrAssBcn phage genomes are similar to Φ CrAss001 but differ from those of other crAssphages.” – specify in what respect they are similar or different.

A/It has been specified in the text. They differ at genomic and protein level.

Line 143. “Functional annotation of all phages by Blastp is presented in Table S3.” Blastp should not be used for functional annotation, since blastp hits might be misannotated. Instead, the curated annotation of Φ CrAss001 might be used.

A/We agree. In fact we used different software for the annotation of phages in the previous version, although we only indicated blastp in the previous version. According to the reviewer suggestion, we have omitted annotation with blastp and instead indicated the different InterPro member databases used (SMART, PFAM, CCED and TIGRFAM) (see new supplementary Data 1 where the annotated genomes are now presented). In addition, we have revised the Φ CrAss001 annotation according with the last version in January 2023 and we have also considered the recently identified proteins by CryoEM, as suggested by another reviewer (Bayfield et al. 2022).

Lines 191-195. “Comparison with 64 non-redundant crAss-like phage genomes in the databases (Fig. S5) showed clear differences. Moreover, relevant differences were also observed between

the phage genomes identified as crAssphages in the databases (Fig.S5), with some shared identities restricted to only one or a few ORFs. This lack of homology raises questions about the criteria used to classify crAssphage, which is discussed later.” – Protein sequence similarity and homology are not the same; establishing homology requires more elaborate steps than creating an all-vs-all similarity matrices. Since the crAssBcn genomes are very similar to CrAss001, and several studies on comparative genomics of crAss-like phages including CrAss001 have already been published, this section should be omitted; the “lack of homology” claim is incorrect.

A/we intended to highlight the fact that the 25 crAssBcn phages did not show similarities with other crAss-like phage genomes. But actually this has already been shown and we agree with the point of the reviewer, therefore this section has been removed for clarity

Lines 208-228 belong to Introduction, not Discussion.

A/We have removed the paragraph and included it in the introduction as requested.

Lines 281-283. “Another unexpected result was the difference in the AAI (< 50 %) of the DNA polymerase I gene of species V and VI compared to the other crAssBcn phages, considering they all belong to the same subfamily and infect the same host.”

A/We believe the reviewer here refers to the same question mentioned above. Please, see our answer above

Line 323. “This brings us to the absence of criteria to define what a crAssphage is.” Again, there are several studies dedicated to this issue.

A/Agreed, with the new taxonomical classification it is now well defined. We have modified the whole paragraph and focused better to our point, that was to indicate that most authors, particularly those attempting to use it as a human fecal marker seem to consider crAssphage as a single virus, while in fact CrAssvirales are a very heterogeneous clade.

Lines 469-471. “In order to recover additional complete viral genomes from the assembly sequences, genomes 20 and 25 were assembled applying the ‘-trusted’ option and employing the Φ CrAss001 genome as a reference.” Please provide the reasoning.

A/ These two genomes are not complete (see table S3) and the “de Novo” strategy was not sufficient. The “Trusted contigs” is the parameter to use an assembly that was previously assembled in a different assembly tool (in our case the CrAss001 phage genome) to provide a guidance during *de novo* assembly. However, it is the parameter for guiding the assembly data input, not for mapping to the genome per se (Prjibelski et al. 2020). It has been clarified in the text.

Lines 472-474. “The genome completeness was checked by checkcomplete and VIBRANT.

A/We agree, the sentence about genome completeness has been removed.

ORF prediction was carried out using Prodigal and functional annotation of predicted genes was performed by Diamond Blastp using the NR NCBI database” Since crAss-like phages have circular genomes, the genome completeness is straightforward: the assembly should be circular. Again, blastp should not be used for functional annotation.

A/As indicated above, other tools were used from the beginning for functional annotation, obtaining the same results as with Blastp. We have now removed blastp annotation and have included the results obtained with different InterPro member databases (SMART, PFAM, CCED and TIGRFAM). In supplementary Data 1 are presented the annotated genomes indicating results with each database. In addition, we have revised the Φ CrAss001 annotation according to the last version in January 2023 and we have also considered the recently identified proteins by CryoEM, as suggested by another reviewer (Bayfield et al. 2022).

A/we are most grateful to the reviewer by the useful and constructive revision. Our study has certainly improved now. We hope we have correctly addressed all the questions

REVIEWER COMMENTS

Reviewer #1 (Remarks to the Author):

I would like to thank the authors for revising and substantially improving their manuscript. I'm glad that my remarks regarding the taxonomic nomenclature and terminology around viral order Crassvirales (crAss-like phages) were mostly taken on board. However, additional improvements to this manuscript are needed.

I would like to stress once again that it is incorrect to call diverse members of the order Crassvirales as "crAssphages". In the same way how various diverse "lambdoid" phages cannot be collectively referred to as "phage lambda". They are all different species, and the terms "phage lambda" or "crAssphage" are reserved to single species each (Lambdavirus lambda and Carjivirus communis). In this connection, I would like to highlight some of the remaining cases of mis-calling crAss-like phages as "crAssphage":

Line 418: "...cultures as hosts to detect crAssphages from wastewater..." refers to crAssBcn phages, which are crAss-like (!) phages, not crAssphage in a strict sense. Please change this to "...cultures as hosts to detect crAss-like phages from wastewater..."

Line 517: "Primers and probes were confirmed as specific for the detection of crAssphages available in genomic databases." The assay reported in this study targets Kehishuvirus primarius (crAss001) and related species. Assays described in Ref. 19 and 23 target the actual crAssphage (Carjivirus communis). Therefore the sentence above should read: "Primers and probes were confirmed as specific for the detection of crAss-like phages available in genomic databases."

Line 575: "Next, crAssphage abundances were calculated..." Since this part refers only to crAssBcn (which are not crAssphage), this should read "Next, crAssBcn abundances were calculated..."

In Figure 1 the coordinate scale on top of the genome alignment is a bit weird (0 coordinate is somewhere in the middle). Also, Φ CrAssBcn25 has unusually small genome is that correct? Half of the terminase gene (first ORF) seems to be missing and the portion of the genome in front of it as well. At the same time, there is a thick pink diagonal block coming from the homologous part of the Φ CrAssBcn17 genome and going in the rightward direction. Is there a large missing part of the Φ CrAssBcn25 genome located to the right of the last large ORF (RNAP subunit)? Adding crAss001 to this comparison would be handy!..

Supplementary Figure 3. Once again, I would like to direct the authors to this preprint: <https://www.researchsquare.com/article/rs-1898492/v1> Some protein functions in this figure

remain mis-identified. For instance, "stabilisation protein" has been shown to be a tail muzzle protein, "transmembrane protein" is a cargo protein, RNA polymerase is mis-spelled etc.

Lines 224-225: Switching between the two types of DNA polymerase at short phylogenetic distances in crAss-like phages has been reported in the literature before (<https://www.nature.com/articles/s41467-021-21350-w>). However, these two types were designated as PolA and PolB by Yutin et al., whereas here the authors claim that both types of polymerases they saw fall into PolA family. I would like to urge the authors to revisit this part and to compare their DNA polymerase variants with those reported by Yutin et al. Is the second type of DNA Pol reported here, the same as PolB reported by Yutin et al.? If that's the case, Fig. 6 needs to be modified accordingly. It would be great if consistent terminology and protein classification was used across different studies...

In my opinion, Fig. 4 and 5 are not essential and can be moved to supplementary materials.

Reviewer #2 (Remarks to the Author):

The manuscript reports on isolation of 25 crAss-like phages, which are actually crAss001-like phages, since they are very similar to thoroughly characterized isolate crAss001. Hence genome analyses of the new phages have little merit per se. Authors failed to investigate the differences between these genomes (no multiple sequence alignments, no phylogenetic trees for DNA polymerase and other key proteins). Phylogenetic trees of the highly variable proteins (constructed with their closest homologs available in GenBank) might shed light on the nature of that variability – was it due to non-orthologous gene replacement, or to accelerated evolution.

The incomplete genomes should be excluded from a comparative genomic study.

Lastly, the sequences being reported should be available for the reviewers at the time of the review.

REVIEWER COMMENTS

Reviewer #1 (Remarks to the Author):

I would like to thank the authors for revising and substantially improving their manuscript. I'm glad that my remarks regarding the taxonomic nomenclature and terminology around viral order Crassvirales (crAss-like phages) were mostly taken on board. However, additional improvements to this manuscript are needed.

I would like to stress once again that it is incorrect to call diverse members of the order Crassvirales as "crAssphages". In the same way how various diverse "lambdoid" phages cannot be collectively referred to as "phage lambda". They are all different species, and the terms "phage lambda" or "crAssphage" are reserved to single species each (Lambdavirus lambda and Carjivirus communis). In this connection, I would like to highlight some of the remaining cases of mis-calling crAss-like phages as "crAssphage":

Line 418: "...cultures as hosts to detect crAssphages from wastewater..." refers to crAssBcn phages, which are crAss-like (!) phages, not crAssphage in a strict sense. Please change this to "...cultures as hosts to detect crAss-like phages from wastewater..."

A/ It has been corrected

Line 517: "Primers and probes were confirmed as specific for the detection of crAssphages available in genomic databases." The assay reported in this study targets Kehishuvirus primarius (crAss001) and related species. Assays described in Ref. 19 and 23 target the actual crAssphage (Carjivirus communis). Therefore the sentence above should read: "Primers and probes were confirmed as specific for the detection of crAss-like phages available in genomic databases."

A/ It has been corrected

Line 575: "Next, crAssphage abundances were calculated..." Since this part refers only to crAssBcn (which are not crAssphage), this should read "Next, crAssBcn abundances were calculated..."

A/ It has been corrected

In Figure 1 the coordinate scale on top of the genome alignment is a bit weird (0 coordinate is somewhere in the middle). Also, Φ CrAssBcn25 has unusually small genome is that correct? Half of the terminase gene (first ORF) seems to be missing and the portion of the genome in front of it as well. At the same time, there is a thick pink diagonal block coming from the homologous part of the Φ CrAssBcn17 genome and going in the rightward direction. Is there a large missing part of the Φ CrAssBcn25 genome located to the right of the last large ORF (RNAP subunit)? Adding crAss001 to this comparison would be handy!..

A/The figure was initially correct, although we agree with the reviewer that it looks weird. This alignment compares each circular genome with the next one looking for synteny between genomes, regardless the first bp of each sequence, that can be different in the different genomes. Therefore, the coordinate scale just informs about where the initial point of each sequence is located. It has been indicated in the figure legend.

About the second part of the question, the reviewer is right, phage Φ CrAssBcn25 is an incomplete genome, showing a shorter sequence as indicated in table S3. We were not sure whether or not to include it in the comparison, so finally and also responding to another reviewer, the genome of Φ CrAssBcn25 have been removed from those figures showing genomes comparison. Instead, in figure 1 we have added crAss001 for a better comparison and according to the reviewer's suggestion. With this modification the alignment looks more consistent and the crAss001 genome is useful as reference. We hope the reviewer will agree with the new version.

Supplementary Figure 3. Once again, I would like to direct the authors to this preprint: <https://www.researchsquare.com/article/rs-1898492/v1> Some protein functions in this figure remain mis-identified. For instance, "stabilisation protein" has been shown to be a tail muzzle protein, "transmembrane protein" is a cargo protein, RNA polymerase is misspelled etc.

A/Unfortunately, as we stuck to the information in databases to build the original figure, this information was not yet available at that moment. We have now updated the annotations and manually indicated the updated functions, identifying them with an asterisk, explained in the figure legend and referenced the study, that has been published only few days ago. Misspelling has been corrected.

Lines 224-225: Switching between the two types of DNA polymerase at short phylogenetic distances in crAss-like phages has been reported in the literature before (<https://www.nature.com/articles/s41467-021-21350-w>). However, these two types were designated as PolA and PolB by Yutin et al., whereas here the authors claim that both types of polymerases they saw fall into PolA family. I would like to urge the authors to revisit this part and to compare their DNA polymerase variants with those reported by Yutin et al. Is the second type of DNA Pol reported here, the same as PolB reported by Yutin et al.? If that's the case, Fig. 6 needs to be modified accordingly. It would be great if consistent terminology and protein classification was used across different studies...

A/ We asked ourselves the same question and checked it previously, comparing the two types of polymerase sequences of crAssBcn phages against all databases and against the polymerase types (polA and PolB) reported by Yutin et al. We confirm that all polymerases in crAssBcn phages belong to PolA family and that they do not correspond to polB reported by Yutin. In fact, according to the phylogenetic analysis depicted in the new figure 5, they derive from PolB. This information was already indicated in the text *"Evolutionary switches in DNA polymerase from type A to B have been described in different families of crAss-like phages⁷, as well as the absence of both enzyme types³¹, but this is not the case for crAssBcn phages, that encoded only a family A DNA polymerase I."*

Moreover, now the reviewer can see the new figure 5 depicting intergenetic comparison in a phylogenetic tree and the location of crAssBcn polymerases. The tree has been built using the sequences reported by Yutin et al. and others in the databases. CrAssBcn polymerases fall in the polA cluster from where the two types of polA diverge. No crAssBcn phages are found in the PolB cluster.

In my opinion, Fig. 4 and 5 are not essential and can be moved to supplementary materials.

A/Former Figure 4 has been moved to supplementary material as requested, Figure 5 have been changed following the request of another reviewer, and a new figure 4 has been built. We hope the reviewer will agree.

A/The authors want to thank the reviewer for devoting efforts revising our manuscript in this second revision round. We hope our answers correctly address all the queries and that the reviewer finds the new version suitable for publication.

Reviewer #2 (Remarks to the Author):

The manuscript reports on isolation of 25 crAss-like phages, which are actually crAss001-like phages, since they are very similar to thoroughly characterized isolate crAss001. Hence genome analyses of the new phages have little merit per se.

Authors failed to investigate the differences between these genomes (no multiple sequence alignments, no phylogenetic trees for DNA polymerase and other key proteins). Phylogenetic trees of the highly variable proteins (constructed with their closest homologs available in GenBank) might shed light on the nature of that variability – was it due to non-orthologous gene replacement, or to accelerated evolution.

A/Actually the multiple sequence alignment of all 24 CrAssBcn phages (we removed Φ CrAssBcn25 as indicated) was done, although we presented only the representative phages of each group in figure 1 because the alignment of all the phages is difficult to visualize in a single figure. Nevertheless, find enclosed now a new, large supplementary figure 3 showing the alignment of all 24 crAssBcn phages.

Moreover, as requested, we have now generated and included 1) alignments of the genes that show differences between phages, and 2) phylogenetic trees of these highly variable proteins, now in a new fig 4 and modification of figure 5. Again, sequences of Φ CrAssBcn25 have been excluded from these analyses. In some figures, we present only the results of the representative phages from the six species, although analyses have been done with all the 24 phages obtaining identical results. Phylogenetic analyses show that some genes (tail protein and endonuclease) clearly derive from a single ancestor and therefore are located in a single cluster while others (tail spike, head, holin and hypothetical protein) are located in different clusters. We are not experts in evolutionary studies, but looking at the bibliography we believe that orthologous gene exchange has occurred.

Similarly phylogenetic tree of the polymerases shows the divergence of the two PolA found in crAssBcn phages that have evolved from a common PolA ancestor, that in turn has evolved from PolB. PolB was included because it was requested by another reviewer. The dichotomy observed and the variable representation of both types in different crAss-like phages derived from different metagenomes and the higher representation of one of the variants of PolA suggest that a non-orthologous gene displacement might finally occur, as it has previously been reported in studies of the polymerase evolution.

The incomplete genomes should be excluded from a comparative genomic study.

A/We have checked again for completeness and circularity of the genomes previously indicated as not complete in supplemental table 3. The genomes of Φ CrAssBcn20, Φ CrAssBcn23 and Φ CrAssBcn25 were designated as not complete because the software used for the analysis indicated that they couldn't be circularized. However, after the revision motivated by the reviewer's request, we observed that while the genome of Φ CrAssBcn25 is clearly incomplete and shows a shorter length, the genomes of Φ CrAssBcn20 and Φ CrAssBcn23 have the same length and ORF number than the rest of crAssBcn phages and their genomes cover the totality of the length of the reference phage genomes including Φ CrAss001, or even are a bit longer (see information in Supplementary table 3). Consequently, while Φ CrAssBcn25 has been removed as requested, Φ CrAssBcn20 and Φ CrAssBcn23 are complete and remain within the comparative study. We have modified the figures, the information in the table and in the text, by removing the information of Φ CrAssBcn25 genome. We have

requested the editor agreement to keep these two phages. We hope the reviewer will agree.

A/ Lastly, the sequences being reported should be available for the reviewers at the time of the review.

A/They are available, we believe the editor has already contacted the reviewer concerning this matter.

A/ The authors would like to sincerely thank the reviewer for the careful review of our study and the concerns raised. To the best of our ability, we have squeezed much more of our data to try to answer them, resulting in more interesting conclusions. We apologize if our knowledge of virus and genetic evolution is not sufficient to properly respond all the reviewer requests. We believe that thanks to the reviewer the work has improved substantially, and we hope that you will find our answers and modifications equally satisfactory.

REVIEWERS' COMMENTS

Reviewer #1 (Remarks to the Author):

I believe all my critical points have now been addressed by the authors. I have no further comments on this manuscript.